# LKB1 coordinates neurite remodeling to drive synapse layer emergence in the outer retina

**Courtney A Burger[1,2], Jonathan Alevy[1,2], Anna K Casasent[1,2], Danye Jiang[1,2], Nicholas E Albrecht[1,2], Justine H Liang[1,2], Arlene A Hirano[3,4], Nicholas C Brecha[3,4], Melanie A Samuel[1,2]***

[1]Department of Neuroscience, Baylor College of Medicine, Houston, United States; [2]Huffington Center on Aging, Baylor College of Medicine, Houston, United States; [3]Department of Neurobiology, David Geffen School of Medicine at UCLA, Los Angeles, United States; [4]United States Veterans Administration Greater Los Angeles Healthcare System, Los Angeles, United States

**Abstract** Structural changes in pre and postsynaptic neurons that accompany synapse formation often temporally and spatially overlap. Thus, it has been difficult to resolve which processes drive patterned connectivity. To overcome this, we use the laminated outer murine retina. We identify the serine/threonine kinase LKB1 as a key driver of synapse layer emergence. The absence of LKB1 in the retina caused a marked mislocalization and delay in synapse layer formation. In parallel, LKB1 modulated postsynaptic horizontal cell refinement and presynaptic photoreceptor axon growth. Mislocalized horizontal cell processes contacted aberrant cone axons in LKB1 mutants. These defects coincided with altered synapse protein organization, and horizontal cell neurites were misdirected to ectopic synapse protein regions. Together, these data suggest that LKB1 instructs the timing and location of connectivity in the outer retina via coordinate regulation of pre and postsynaptic neuron structure and the localization of synapse-associated proteins.

*For correspondence:
msamuel@bcm.edu

Competing interests: The authors declare that no competing interests exist.

## Introduction

The precise spatial and temporal regulation of neuron maturation and synapse formation is crucial to ensure the fidelity of neural circuits. Many genes have been identified that regulate the initial steps of neuron maturation, including neuron fate and neurite development (*Bassett and Wallace, 2012*; *Blackshaw et al., 2004*; *Sainath and Gallo, 2015*). However, we know little about how these processes contribute to the emergence of ordered connectivity. In part, this is because many neurons form a large number of synapses, and these synapses are broadly distributed along the neuron. This makes it difficult to resolve the cellular and molecular drivers of these events. To solve this problem, we use the outer retina since it is one of the few regions in the mammalian central nervous system (CNS) where neuronal cell types and their basic connectivity are known (*Sanes and Zipursky, 2010*; *Behrens et al., 2016*; *Shekhar et al., 2016*). In adult animals, outer retina synapses are localized in a thin band known as the outer plexiform layer (OPL) which consists of sublamina comprised of cone and rod synapses. Because each photoreceptor forms connections at one large distal location, the relationship between the structure and maturation of both pre and postsynaptic neurons relative to their connectivity can be directly examined.

OPL development begins after birth and involves interactions between four neuron types: presynaptic rods and cones, and postsynaptic bipolar and horizontal cells (*Cepko, 2014*). Cones and horizontal cells are the earliest born neurons in this circuit (~E12-E17), and they are the first to form contacts in the emerging OPL. Newborn cones extend axons (P0-P5), and in concert, horizontal cells

restrict their arbors to form the horizontal structure for which they are named. These contacts coincide with displacement of nuclei from the cell-free nascent OPL, which is visible by P5. In parallel, synapse-specific proteins are trafficked to cell terminals, corresponding with the onset of synapse formation. OPL sublamination of rod and cone synapses begins at P14, and the OPL is mature by P30 (*Figure 1A*, *Olney, 1968*; *Blanks et al., 1974*; *Rich et al., 1997*; *Sarin et al., 2018*). These events generally mirror those of other laminated CNS regions, such as the cerebellum, where neurons remodel together to give rise to ordered connectivity (*Miterko et al., 2018*). However, in the outer retina, as in the brain, fundamental questions remain: 1) what are the pathways that instruct synapse layer emergence; 2) how does the structure of each neural partner impact the development and connectivity of the other; and 3) how does neurite patterning influence the localization of nascent synapse proteins that may instruct ordered connectivity?

To begin to resolve these questions, we focused on the serine/threonine kinase LKB1 (Liver Kinase B1, also called STK11 or Par4; encoded by *Stk11*). LKB1 regulates 14 kinases of the AMPK subfamily (AMPKα1/α2, SAD-A/B, NUAK1/2, SIK1-3, MARK1-4 and SNRK, *Lizcano et al., 2004*; *Jaleel et al., 2005*) and has cell-specific roles in polarity, neuron maturation, and axon formation (*Kuwako and Okano, 2018*; *Shelly et al., 2007*; *Barnes et al., 2007*; *Courchet et al., 2013*). However, it is unknown whether or how LKB1-driven axon emergence impacts synapse localization. Given its high levels of expression in the retina (*Samuel et al., 2014*), we asked whether LKB1 may participate in programing outer retina connectivity and if these alterations might inform the cellular and molecular processes that drive synapse layer emergence. We show that loss of LKB1 delays formation of the OPL. This defect is accompanied by abnormal horizontal cell neurite restriction, with apical and basal processes that extend beyond the nascent synapse region. In concert, LKB1 mutants showed abnormal cone axon extension, and mislocalized axons were contacted by aberrant horizontal cell processes. Cone axon extension deficits were accompanied by specific defects in synaptic protein localization. In particular, we found that RIBEYE and VGLUT1 accumulated in the somas of LKB1 mutant cones and failed to properly reach the axon terminal. Misplaced synapse proteins were contacted by ectopic horizontal cell neurites. Together, these data suggest a model in which LKB1-mediated presynaptic axon extension and postsynaptic neurite refinement are coordinately required for precisely refining neuron structure and synapse protein localization.

## Results

### LKB1 regulates the timing of synapse layer emergence

In control animals the nascent OPL first appears as small discontinuous patches at postnatal day 3 (P3), where horizontal cell neurites and cone terminals begin to form contacts. These patches begin to align and exclude the dense nuclei that populate the outer retina at P5. A single cell-free OPL layer forms as patches fully converge by P8 and the cellular boundaries of the ONL and INL become defined (*Figure 1A*). Correspondingly, we found that the levels of *Stk11* mRNA are highest in early development at P5 when synapses begin to emerge (*Figure 1—figure supplement 1*), with expression present in both inner and outer retina. To determine the role of LKB1 in the emergence of synaptic connectivity we generated full retina LKB1 knockout mice using the conditional allele $Stk11^{F/F}$ (previously called $Lkb1^{F/F}$, *Bardeesy et al., 2002*) and the *Vsx2-cre* line (previously called *Chx10-cre*, *Rowan and Cepko, 2004*) that expresses *Cre* in embryonic retinal progenitors to generate $Stk11^{F/F}$; *Vsx2-Cre* animals. This line is hereafter referred to as Lkb1-Ret. Defects in LKB1 mutant retinas became apparent as the synapse layer began to emerge. While control animals displayed nuclei-free patches at P3 that are localized 39.1 ± 0.3 µm away from the apical side of the outer retina, in Lkb1-Ret mice OPL patches were small and difficult to visualize (*Figure 1B*), displaced closer to the apical retinal surface relative to control mice (29.6 ± 0.4 µm away, *p*<0.0001), and reduced in number (40.1% reduction, *p*=0.0286, *Figure 1C*). At P5, these patches converged in control animals to form a single layer 54.1 ± 0.2 µm away from the apical surface of the retina, forming the OPL (*Figure 1B, D*). In contrast, the outer retina synapse layer was largely absent from Lkb1-Ret animals at this time point, appearing instead as discontinuous regions that were misaligned and located closer to the apical retina surface (36.2 ± 0.3 µm away, *p*<0.0001, *Figure 1B, D*). These defects were accompanied by a 56.0% reduction in OPL area (*p*=0.004; *Figure 1E*).

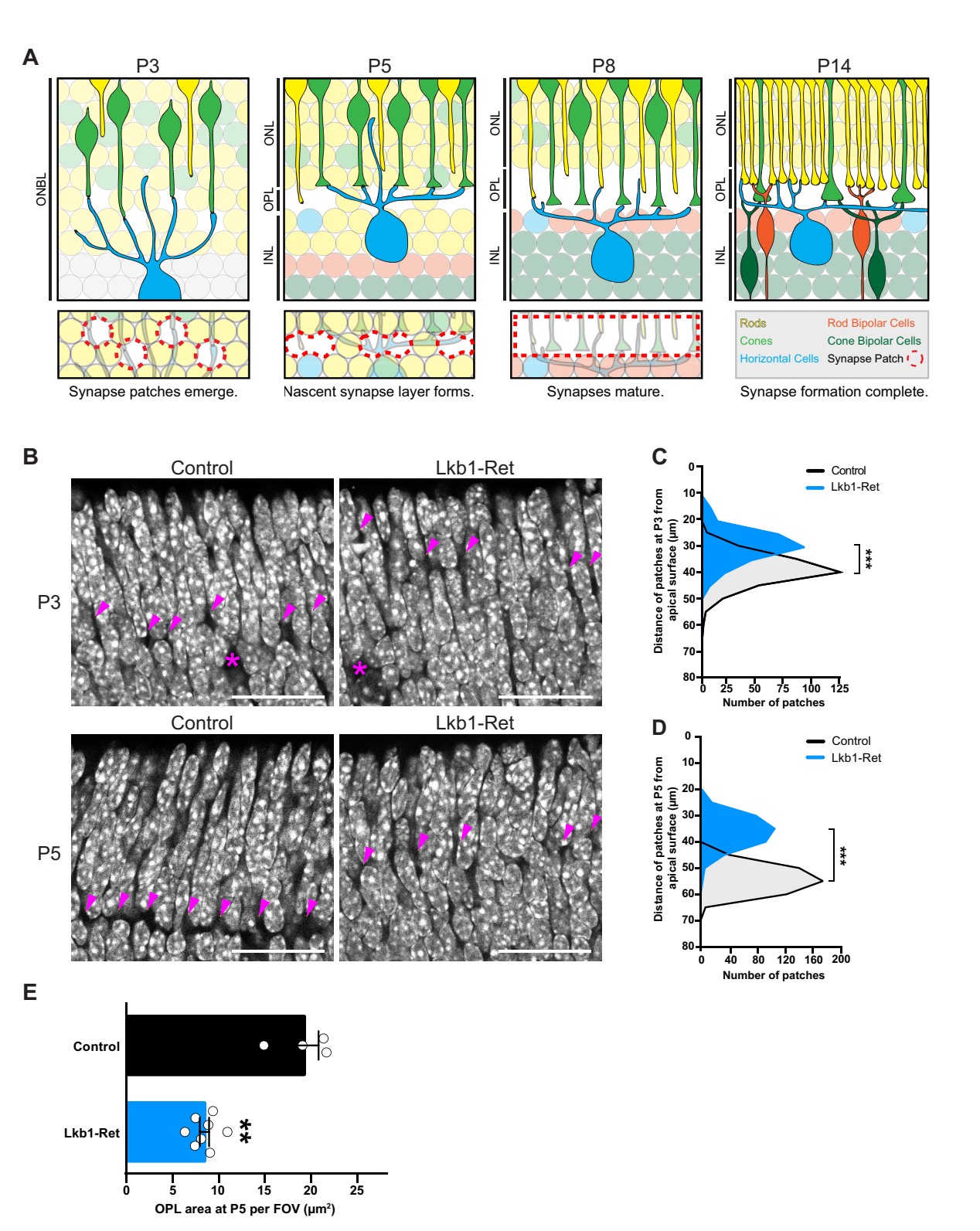

**Figure 1.** LKB1 is required for outer plexiform layer formation. (**A**) Schematic of outer plexiform layer (OPL) development from P3 to P14. The outer retina contains developing rods (yellow) and cones (green) that extend axons beginning at P1. Cones form contacts with horizontal cell interneurons (blue) at P3, where nascent synapse patches emerge at sites of contact (red circles), followed by synaptogenesis beginning at P5. This corresponds with OPL formation. Rods begin to form synaptic connections with horizontal cells by P8, and bipolar cells also begin to become integrated at this time

*Figure 1 continued on next page*

*Figure 1 continued*

(orange, rod bipolars; dark green, cone bipolars). Ribbon synapse formation is complete and OPL sublamination begins by P14. (ONBL, outer neuroblast layer; ONL, outer nuclear layer; OPL, outer plexiform layer; INL, inner nuclear layer). (**B–D**) In control animals, the nascent OPL is first visible at P3 when it appears as small discontinuous gaps in the nuclear staining (DAPI, grey; arrowheads point to patches). These patches are distinct from gaps caused by horizontal cell bodies (magenta stars). In Lkb1-Ret animals, these OPL patches are small and located closer to the apical surface (**B**) and are reduced in number (**C**, n = 322 control cells and n = 252 Lkb1-Ret cells). N = 4 control and N = 4 Lkb1-Ret animals. At P5, nascent patches converge in control animals to generate a single continuous layer, forming the OPL. In Lkb1-Ret animals, the OPL was discontinuous and patches were misaligned and located closer to the apical surface (**D** n = 487 control cells and n = 339 Lkb1-Ret cells). This resulted in a marked decreased in total OPL area (**E**). N = 4 control and N = 8 Lkb1-Ret animals. Scale bars = 25 μm. Data are represented as a distribution of the distance of patches from the apical surface (**B,D**, ***p<0.001, unpaired two-tailed Student's *t* test) or as the mean ± the s.e.m. (**E**, **p<0.01, non-parametric Mann-Whitney Rank Sum U-test).

The online version of this article includes the following figure supplement(s) for figure 1:

**Figure supplement 1.** *Stk11* is highly expressed throughout the retina in early development.
**Figure supplement 2.** AMPK does not regulate outer retina development.

We next examined OPL segregation, which occurs as nuclei become excluded from the synaptic layer and migrate into the outer and inner nuclear layers. To define the borders of the OPL, we first stained with DAPI to visualize nuclei. The OPL is precisely segregated at P8 in control animals, and nuclei are absent from the OPL (*Figure 2A*). In contrast, Lkb1-Ret animals display mislocalized nuclei within the OPL, with an 85.6% increase in ectopic nuclei (p=0.0294, *Figure 2A–B*). To visualize how these defects impacted neuron segregation, we labeled rods and bipolar cells whose nuclei reside at the upper and lower OPL boundaries. In control animals, cellular boundaries are clearly defined with rods appropriately localized to the ONL and bipolars localized to the INL (*Figure 2C*). In contrast, Lkb1-Ret animals displayed defects in rod neuron segregation. This resulted in some rod nuclei remaining below the nascent OPL and a 57.8% reduction in total OPL area (p=0.0286, *Figure 2C–D*). These alterations do not appear to be due to differences in neuron numbers, as both pre and postsynaptic outer retina neuron populations are present in equal numbers in LKB1 mutants (*Figure 2—figure supplement 1*). These data indicate that LKB1 coordinately regulates OPL emergence and nuclear segregation.

We next asked whether LKB1 mediated outer retina synapse emergence involves similar pathways to those in synapse decline. We previously showed in old age that AMPK acts downstream of LKB1 to regulate rod synapse maintenance (*Samuel et al., 2014*). To test the role of AMPK in development, we generated animals in which both alpha subunits of *Ampk* (*Prkaa1 and Prkaa2*) were specifically deleted in retina (*Prkaa1$^{F/F}$Prkaa2$^{F/F}$; Vsx2-Cre*, *Nakada et al., 2010*, hereafter referred to as Ampk-Ret). AMPK was dispensable for outer retina synapse emergence: Ampk-Ret mice showed no observable defects in OPL emergence or organization at P5, and the OPL was present at the proper location and time (*Figure 1—figure supplement 2*). In addition, outer retina neurons were morphologically normal at this time point (*Figure 1—figure supplement 2*). Thus, the downstream drivers of outer retina synapse emergence differ from those involved in synapse maintenance and aging.

## LKB1 regulates horizontal cell neurite refinement

Horizontal cells and cones are the first to arrive and form contacts in the nascent OPL in a process that involves extensive neurite remodeling (*Sarin et al., 2018*; *Huckfeldt et al., 2009*; *Barrasso et al., 2018*). We thus examined the morphology of these neurons (*Figure 3* and *Figure 3—figure supplement 1*). In control animals, horizontal cells first emerge as radial neurons (*Schubert et al., 2010*), exhibiting long apical and basal neurites (*Figure 3A*). Between P3 and P5, horizontal cells then undergo a process of neurite refinement in which apically and basally targeted neurites become lateralized through largely unknown mechanisms to form a horizontal neural structure (*Poché and Reese, 2009*). At P3, Lkb1-Ret horizontal cells were structurally indistinguishable from control animals (p≥0.12, *Figure 3A–B* and *Figure 3—figure supplement 1A–B*). However, by P5 Lkb1-Ret horizontal cells displayed marked defects, extending numerous, long branches into the outer and inner retina (*Figure 3C* and *Figure 3—figure supplement 1C–D*). This resulted in horizontal cell processes spanning a significantly increased retina area relative to controls (p<0.05, *Figure 3D*). Horizontal cell neurites became largely lateralized in Lkb1-Ret animals by P8, although occasional, long apical horizontal neurites persisted at this timepoint (*Figure 3E–F*).

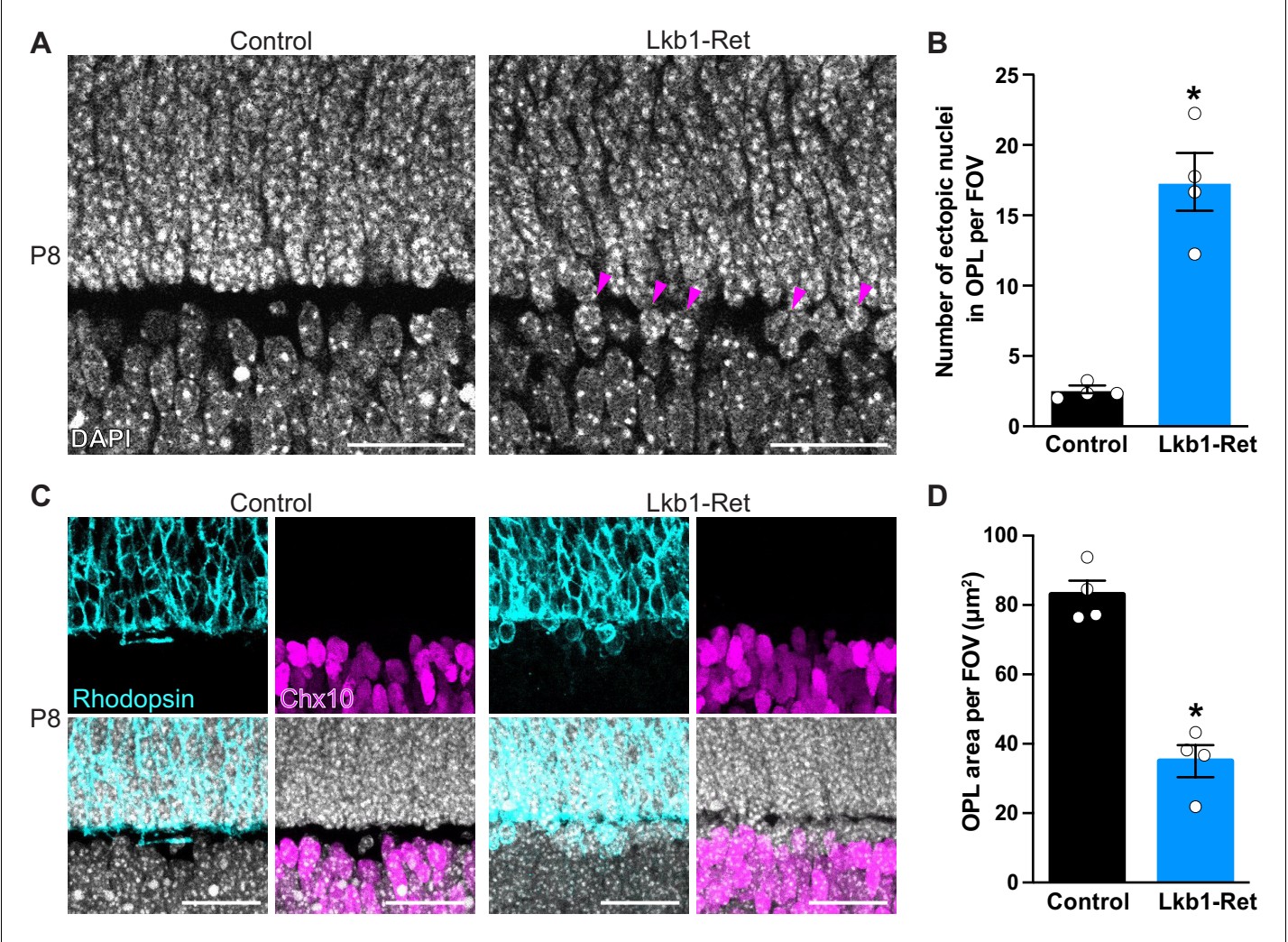

**Figure 2.** The outer plexiform layer is disorganized in LKB1 mutant animals. (A-B) The OPL was visualized in Lkb1-Ret animals and control littermates at P8 following staining with DAPI. The OPL of Lkb1-Ret animals contains ectopic nuclei (arrows, **A**) that were increased in number relative to controls (**B**). (C–D) Cellular segregation was visualized using antibodies that demark the upper (rods, rhodopsin, blue) and lower (bipolar cells, Chx10, magenta) OPL boundaries. Lkb1-Ret animals displayed defects in neuron segregation with some rod nuclei remaining below the nascent OPL (**C**). The total area of the OPL (**D**) was also decreased in Lkb1-Ret animals relative to littermate controls. N = 4 control and N = 4 Lkb1-Ret animals. Scale bar = 25 μm. Data are represented as the mean ± the s.e.m. *p<0.05, non-parametric Mann-Whitney Rank Sum U-test.
The online version of this article includes the following figure supplement(s) for figure 2:

**Figure supplement 1.** Outer retina neuron cell numbers are normal in LKB1 mutant animals.

We next investigated whether the defects in horizontal cell refinement represented a cell-intrinsic role for LKB1 in shaping horizontal cell architecture. To examine this, we selectively deleted *Stk11* from horizontal cells early in development (P2, *Barrasso et al., 2018*) using a transgenic line that expresses *Cre* only in these neurons (*Gja10-ires-iCre*, previously called *Cx57-ires-iCre*, *Hirano et al., 2016*) to generate *Stk11^F/F^;Gja10-ires-iCre* animals. This line is hereafter referred to as Lkb1-HC. In situ for *Stk11* in Lkb1-HC retina confirmed cell-specific deletion of this transcript in horizontal cells (*Figure 4—figure supplement 1*). Lkb1-HC mice showed no observable defects in OPL emergence or organization at P5, and the OPL was present at the proper location and time (*Figure 4A–C*). In addition, horizontal cells were morphologically normal at this time point (*Figure 4D*), and horizontal cells properly refined their neurites to the OPL (*Figure 4E*). These data indicate that LKB1 is dispensable in horizontal cells for the refinement of their arbors and for OPL emergence.

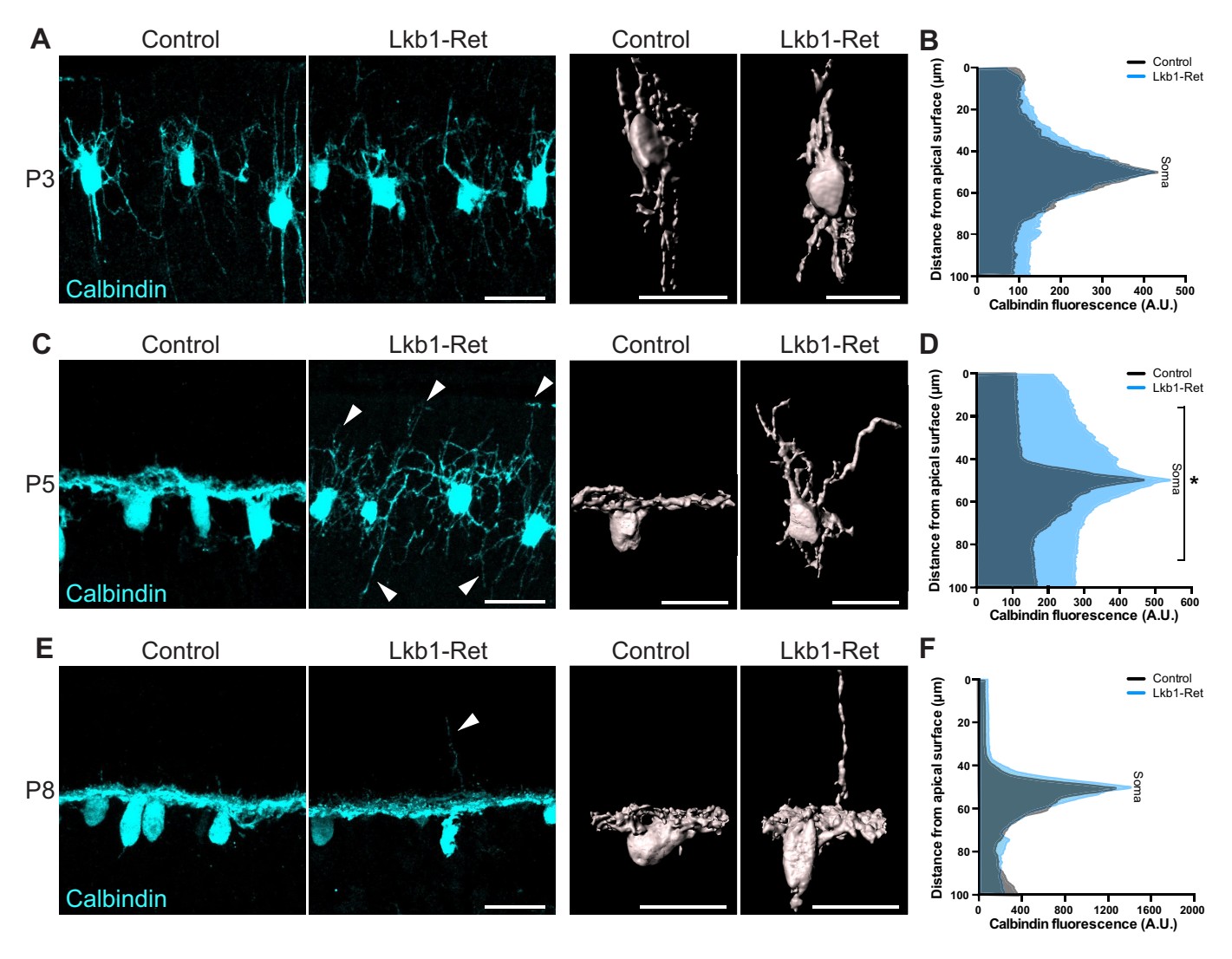

**Figure 3.** Horizontal cell neurite restriction is altered with loss of LKB1. Horizontal cells and their neurites were identified in Lkb1-Ret and littermate controls during postnatal development using an antibody to calbindin (cyan). (A–B) Representative and reconstructed images (A) and quantification (B) of horizontal cell morphology at P3. Horizontal cells in both Lkb1-Ret and control animals exhibit stellate morphologies, and no significant structural differences were observed. N = 3 littermate controls and N = 7 Lkb1-Ret animals. (C–D) Representative and reconstructed images (C) and quantification (D) of horizontal cell morphology at P5. Lkb1-Ret horizontal cells fail to restrict their arbors at P5 (arrows) and instead display a marked radial morphology that results in horizontal processes spanning a significantly increased retinal area. N = 4 control and N = 8 Lkb1-Ret animals. (E–F) Representative and reconstructed images (E) and quantification (F) of horizontal cell morphology at P8. Horizontal cells in Lkb1-Ret animals refined their arbors, though occasional extensions into the outer retina were observed (arrow). N = 4 control and N = 4 Lkb1-Ret animals. Scale bars = 25 µm. Data are represented as the mean fluorescence relative to the distance from the apical surface. *p<0.05, unpaired two-tailed Student's *t* test.

The online version of this article includes the following figure supplement(s) for figure 3:

**Figure supplement 1.** Horizontal cells fail to restrict their neurites at the appropriate developmental time.

## Cone axon extension and maturation are disrupted in LKB1 mutant mice

What might be responsible for the defects in horizontal cell refinement? Given the high levels of LKB1 in outer retina neurons at P5, we questioned whether horizontal cell presynaptic partners might play a role in the horizontal cell defects we observed. In wild type animals, horizontal cells exclusively form contacts with developing presynaptic cone axons from P3-P5 as horizontal cells lateralize, while rod contacts occur later (>P8, *Bassett and Wallace, 2012*). We thus focused our

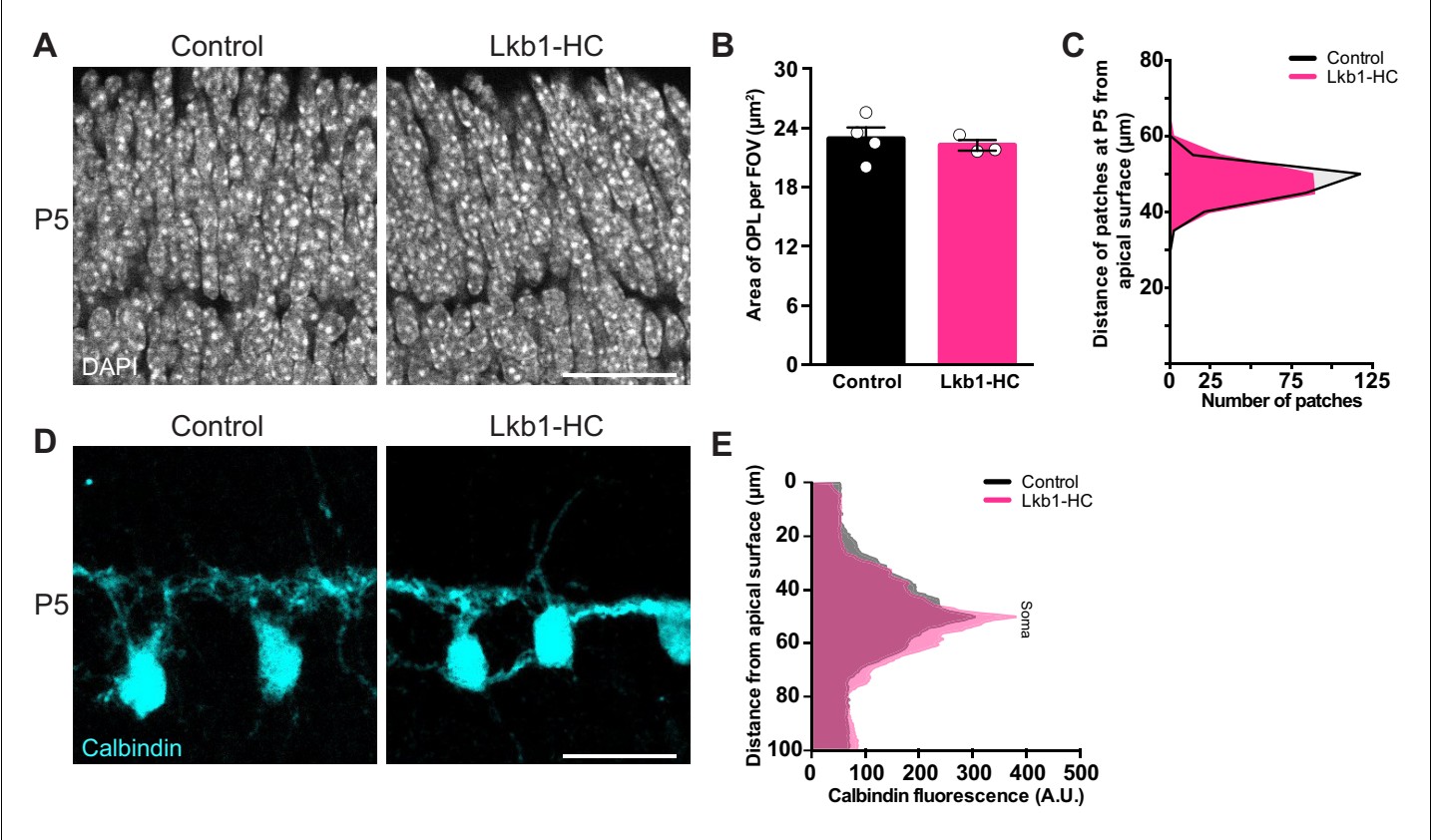

**Figure 4.** LKB1 is not required in horizontal cells to restrict their neurites. Outer retina emergence and horizontal cell restriction were visualized in Lkb1-HC and littermate controls during postnatal development at P5. (**A–C**) Representative images (**A**) and quantification (**B**) of OPL emergence (DAPI, grey) and (**C**) distance of OPL patches from the apical surface at P5 in Lkb1-HC and littermate controls. The OPL emerges in Lkb1-HC animals at the proper time and location (**B**) and is located the same distance from the apical surface as controls (**C**, n = 239 control cells and n = 235 Lkb1-HC cells). N = 4 littermate controls and N = 3 Lkb1-HC animals. (**D–E**) Representative images (**D**) and quantification (**E**) of horizontal cell (calbindin, cyan) morphology at P5. Lkb1-HC horizontal cells restrict their arbors, spanning the same area as control mice. N = 4 control and N = 3 Lkb1-HC animals. Scale bars = 25 μm. Data are represented as the mean ± the s.e.m. (**B**, p>0.05, non-parametric Mann-Whitney Rank Sum U-test), as a distribution of the distance of patches from the apical surface (**C**, p>0.05, unpaired two-tailed Student's *t* test), or as the mean fluorescence relative to the distance from the apical surface (**E**, p>0.05, unpaired two-tailed Student's *t* test).

The online version of this article includes the following figure supplement(s) for figure 4:

**Figure supplement 1.** *Stk11* is lost in horizontal cells in Lkb1-HC animals.

attention on cones. To examine these cells, we used cone-specific antibodies to resolve the positioning of cone axons (*Table 1*). In control mice, cone axon extension was present at P1, and by P3, cones had reached 70.4% of their terminal axon length (*Figure 5A*). Axon extension was complete by P8 when cones reached their terminal axon length (34.7 ± 0.6 μm). Lkb1-Ret cones showed marked defects in cone axon extension. Mutant cones failed to properly develop axons at P1. Instead, axon extension was reduced, resulting in a decrease in average terminal axon length throughout the first postnatal week (68.3%, 57.7%, and 56.0% reduction in axon length relative to controls at P1, P3, and P5, p≤0.02, *Figure 5A–E*). Moreover, a large number of Lkb1-Ret cones lacked axons at early time points (*Figure 5C–E,G*). By P8, the majority of Lkb1-Ret cones had extended axons (*Figure 5F,G*), but they remained shorter than control cones (83.5% of the length of controls), and a small fraction still failed to reach the OPL (3.5%, p=0.0286, *Figure 5H*). Together, these data indicate that LKB1 is coordinately required for cone axon extension and restriction of horizontal cell process to enable OPL emergence.

We then examined the contacts between horizontal cells and cones. We reasoned that if cone and horizontal cell interaction modulates synapse layer emergence, then: 1) normal cone axon extension should be accompanied by a coordinated reduction in horizontal cell neurite length, and 2) the

**Table 1.** Antibodies used in LKB1 mutant tissue analysis.
Antibodies were utilized that label individual neuron populations and synapses in the outer retina.

| Antibody name | Immunogen | Labeling specificity | Source | Concentration |
|---|---|---|---|---|
| Calbindin D-28k | Full-length recombinant human Calbindin D-28K | Horizontal cells; amacrine cells; retinal ganglion cells | Novus biologicals; chicken polyclonal; NBP2-50028; no RRID | 1:2000 |
| Calbindin D-28k | Recombinant rat calbindin D-28k (CB) | Horizontal cells; amacrine cells; retinal ganglion cells | Swant; rabbit polyclonal; CB38; RRID:AB_10000340 | 1:10,000 |
| Chx10 | Recombinant protein derived from the N terminal of the human Chx10 protein conjugated to KLH (aa 1–131) | Bipolar cells | Exalpha; sheep polyclonal; X1180P; RRID:AB_2314191 | 1:300 |
| OPN1SW | Peptide mapping at the N-terminus of the opsin protein encoded by OPN1SW of human origin | Cone photoreceptors | Santa Cruz; goat polyclonal; sc-14363; RRID:AB_2158332 | 1:500 |
| Piccolo | Recombinant protein corresponding to AA 4439 to 4776 from rat Piccolo | Ribbon synapses | Synaptic Systems; rabbit polyclonal; 142 003; RRID:AB_2160182 | 1:500 |
| Protein Kinase C alpha (PKCa) | Purified bovine brain protein kinase C | Rod bipolar cells | Abcam; mouse monoclonal; ab31; RRID:AB_303507 | 1:500 |
| PSD95 | Synthetic peptide corresponding to Mouse PSD95 aa 1–100 (C-terminal) conjugated to keyhole limpet haemocyanin. | Photoreceptor terminals | Abcam; goat polyclonal; ab12093; RRID:AB_298846 | 1:500 |
| Rhodopsin | Recombinant fragment corresponding to Bovine Rhodopsin (N terminal) | Rod photoreceptors | Abcam; mouse monoclonal; ab98887; RRID:AB_10696805 | 1:500 |
| RIBEYE | Recombinant protein corresponding to AA 95 to 207 from rat Ribeye | Ribbon synapses | Synaptic Systems; rabbit polyclonal; 192 103; RRID:AB_2086775 | 1:500 |
| VGLUT1 | Recombinant protein corresponding to AA 456 to 560 from rat VGLUT1 | Photoreceptor terminals | Synaptic Systems; rabbit polyclonal; 135 302; RRID:AB_887877 | 1:500 |

axon extension deficits in LKB1 mutant cones should be followed by mislocalized contacts between horizontal cells and cones. To examine this, we co-labeled cones and horizontal cells and quantified the number of terminal contacts over time. We began our analysis in control animals at P3 when cone axons are 70.4% of their adult length and horizontal cells have yet to restrict their arbors to the OPL (*Figure 3A*; *Figure 5A*). Many horizontal cell processes were apposed to cone terminals at this time (77.8%), and terminal contacts occurred within 20.4 ± 1.4 µm of the horizontal cell soma (*Figure 6A–C*). In addition, horizontal cell contacts with cones outside of the axon terminal occurred with some frequency in controls, with 22.2% of contacts occurring at non-terminal positions. By P5, however, the fidelity and restriction of these appositions to the axon terminal increased to nearly 96.8% in controls. Their location reflected horizontal neurite restriction, with an average contact distance within 2.6 ± 0.1 µm of the horizontal cell soma (*Figure 6D–F*). At this time point and beyond, horizontal cell contacts on control cones rarely occurred outside of the axon terminal (3.2% at P5 and 4.1% at P8, *Figure 6E,H*).

In Lkb1-Ret mice we observed three notable differences in this pattern. First, the relative frequency of horizontal cell contacts with cones differed, with more occurring at non-terminal positions. Second, the presence of these alterations corresponded with the onset of horizontal cell restriction defects. Third, the location of pre and postsynaptic contacts was corrected in concert with cone axon extension. In particular, at P3 when control and Lkb1-Ret horizontal cells are morphologically

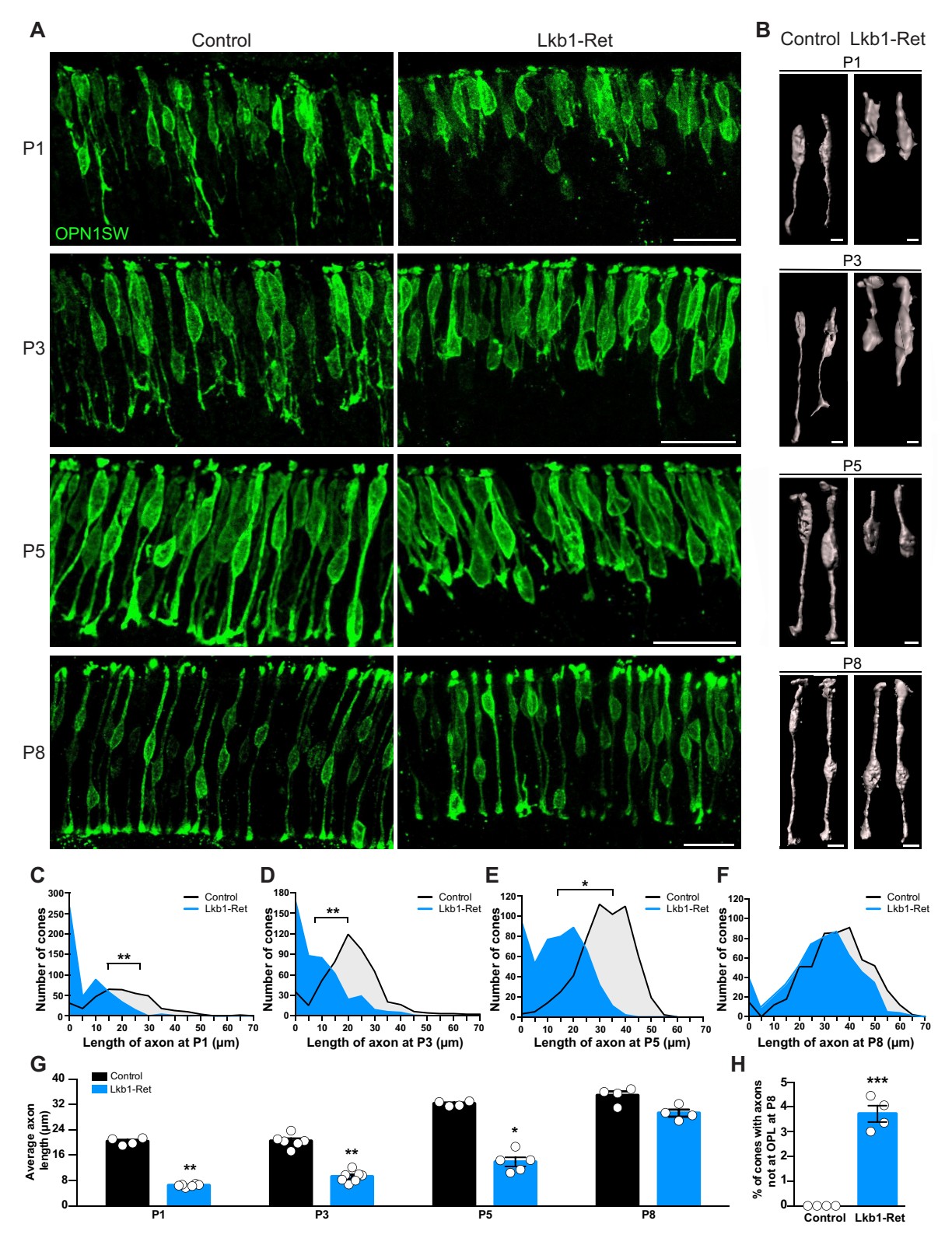

**Figure 5.** Cone axon extension requires LKB1. Cones and their axons were identified in Lkb1-Ret animals and littermate controls during postnatal development using an antibody to OPN1SW (green). (A–B) Representative images (A) and reconstructed cones (B) over development. (C–F) The distribution of the length of cone axons in control and Lkb1-Ret animals was quantified at P1 (C, n = 376 control cells and n = 515 Lkb1-Ret cells), P3 (D, n = 488 control cells and n = 478 Lkb1-Ret cells), and P5 (E, n = 572 control cells and n = 505 Lkb1-Ret cells). At each time point, Lkb1-Ret cones were

*Figure 5 continued on next page*

Figure 5 continued

shorter, and many also lacked axons. By P8, the distribution of length of cone axons in control and Lkb1-Ret animals was normal (**F**, n = 560 control cells and n = 552 Lkb1-Ret cells). (**G**) Quantification of cone axons over development. Lkb1-Ret animals displayed defects in cone axon extension beginning at P1 that persisted at P3 and P5, resulting in a significant decrease in axon length (N ≥ 4 control and N ≥ 5 Lkb1-Ret animals). (**H**) OPL localization of cone terminals was quantified at P8. There was a significant increase in the number of cones that failed to reach the OPL (N = 4 control and N = 4 Lkb1-Ret). Scale bars = 25 μm (**A**) and 5 μm (**B**). Data are represented as the mean ± the s.e.m. (**G–H**) or as the distribution of the length of cone axons (**C–F**). \*\*\*p<0.001, \*\*p<0.01, \*p<0.05, non-parametric Mann-Whitney Rank Sum U-test.

similar (*Figure 3A*), 66.2% of horizontal cell processes were apposed to cone terminals at this time, and these occurred within 22.0 ± 0.6 μm of horizontal cell soma. However, an increase in the number of non-axon contacts was observed in LKB1 mutants relative to controls (34.3% increase, *Figure 6A–C*). Furthermore, at P5, the location of Lkb1-Ret horizontal cell contacts differed significantly: the majority were 16.7 ± 0.8 μm away from the horizontal cell soma (an 84.4% increase in distance), consistent with the defects in cone axon extension and horizontal cell neurite refinement at this time (84.5% increase in horizontal cell neurite length, p=0.0159, *Figure 6D–F*). In addition, horizontal cells made more contacts with cones beyond the axon terminal (82.1% increase in non-terminal contacts), with many occurring at the soma and the cone inner and outer segment (*Figure 6E*). These contact location errors were largely corrected by P8, as Lkb1-Ret cone axons grew (*Figure 6G–I*). Together, these data suggest cone terminal localization, horizontal neurite restriction, and contact location may be coordinately regulated.

## LKB1 is required for synapse-associated protein localization in cones

A key feature of laminated circuits is that synapse formation occurs at a restricted cellular location. This restriction is vital for circuit organization and thus function. What determines the location of these contacts? While several factors are likely to play roles, synapse-associated proteins have been shown to be required for OPL organization (*Matsuoka et al., 2012*; *Li et al., 2015*). Thus, we wondered how OPL emergence might impact the presence and localization of synapse-associated molecules. To assess this, we defined the timing, levels, and location of four cone terminal proteins to identify those present at the early stages of OPL formation (*Table 1*). In control animals, the synapse-associated proteins RIBEYE, piccolo, and PSD95 were absent prior to synapse formation but present and restricted to the OPL at P5, when synapses emerge (*Figure 7A*). The levels and localization of these proteins increased at P8 as the OPL matured (*Figure 7B*). This pattern of synapse protein location differed in LKB1 mutants. Consistent with the absence of the OPL at P5, the levels of synapse-associated proteins were reduced in LKB1 mutants at P5 (*Figure 7A–C*). As the OPL emerged in LKB1 mutants at P8, synapse proteins became present, though their levels were lower than those in control animals (*Figure 7B–C*).

To examine LKB1-driven synapse protein disorganization in more detail, we obtained high-resolution views of RIBEYE localization in control and Lkb1-Ret cones at P5 using expansion microscopy (*Figure 8—figure supplement 1*; *Chen et al., 2015*). In control animals, all cone terminals displayed one or more RIBEYE puncta (*Figure 8A–B*), consistent with the numerous ribbon synapses formed by cone pedicles (*Mercer and Thoreson, 2011*). Further, RIBEYE was largely restricted to the cone terminal. We noted two differences in RIBEYE localization in LKB1 mutant cones. First, RIBEYE was absent from cone terminals in some cases (*Figure 8B*, star). Second, RIBEYE was present at higher levels in mutant cone somas (*Figure 8A–B*, arrowhead). These data suggest outer retina synapse emergence defects are accompanied by alterations to the localization of synapse-associated proteins.

## LKB1 is required for early localization of VGLUT1

We next considered the timing of synapse layer emergence relative to the presence of synapse-associated proteins. Interestingly, VGLUT1 was present in cones earlier than other synapse-associated proteins: it could be visualized as early as P1 in controls, preceding other synapse-associated proteins by at least 48 hr (*Figure 7*). In control animals at P1, VGLUT1 was predominantly found in cones in the latter third of the maturing axon and at the axon terminal (*Figure 9A*). By P5, VGLUT1 was largely restricted to cone axon terminals (99%, *Figure 9B*), and VGLUT1 staining became more pronounced from P5 to P8 where it overlapped with cone pedicles as they grew (*Figure 9C*).

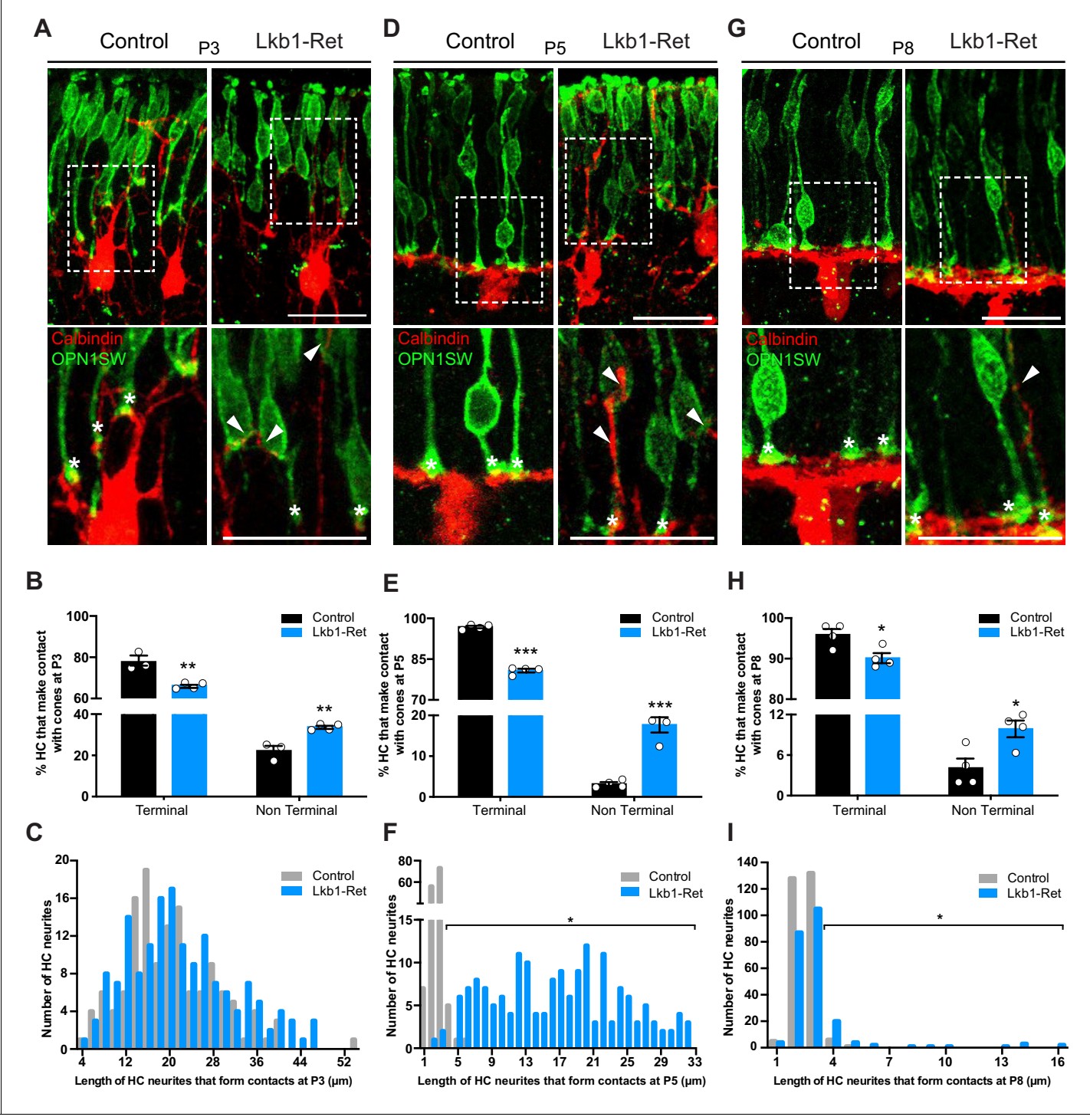

**Figure 6.** Horizontal cells in LKB1 mutants contact cones at terminal and non-terminal positions. Horizontal cell and cone contacts (calbindin, red; OPN1SW, green) were identified in Lkb1-Ret and littermate controls during postnatal development. (**A**) Representative images of horizontal cell-cone contacts at P3. Higher magnification images are displayed of the contacts (stars demarcate terminal contacts; arrows demarcate non-terminal contacts). Scale bars = 25 μm. (**B–C**) The location of contacts between cones and horizontal cells was quantified at P3. Lkb1-Ret animals showed a small but significant reduction in terminal contacts and an increase in non-terminal contacts relative to controls (**B**). The length of horizontal cell neurites that formed contacts with cones in control and Lkb1-Ret animals were measured and binned. The mean distance of the terminal contacts relative to the location of horizontal cell somas did not significantly differ between groups (**C**, n = 135 control cells and n = 159 Lkb1-Ret). N = 3 control and N = 5 Lkb1-Ret animals. (**D**) Representative images of horizontal cell-cone contacts at P5. Higher magnification images are displayed of the contacts (stars demarcate terminal contacts; arrows demarcate non-terminal contacts). (**E–F**) The location of contacts between cones and horizontal cells was

*Figure 6 continued on next page*

*Figure 6 continued*

quantified at P5. Lkb1-Ret animals showed a significant reduction in terminal contacts and an increase in non-terminal contacts relative to controls (E). The length of horizontal cell neurites that formed contacts with cones in control and Lkb1-Ret animals were measured and binned. The mean distance of the terminal contacts relative to the location of horizontal cell somas significantly increased in Lkb1-Ret animals (F). N = 4 control and N = 5 Lkb1-Ret animals. (G) Representative images of horizontal cell-cone contacts at P8. Higher magnification images are displayed of the contacts (stars demarcate terminal contacts; arrows demarcate non terminal contacts). (H–I) The location of contacts between cones and horizontal cells was quantified at P8. Lkb1-Ret animals showed a small but significant reduction in terminal contacts and an increase in non-terminal contacts relative to controls (H). The length of horizontal cell neurites that formed contacts with cones in control and Lkb1-Ret animals were measured and binned. The mean distance of the terminal contacts relative to the location of horizontal cell somas significantly increased in Lkb1-Ret animals (I). N = 4 control and N = 4 Lkb1-Ret animals. Scale bars = 25 μm. Data are represented as the mean ± the s.e.m. (B, E, H, ***p<0.001, **p<0.01, *p<0.05, unpaired t-test across rows corrected for multiple comparisons using the Holm-Sidak method) or as the distribution of length of horizontal cell neurites that form contacts with cones (C, F, I, *p<0.05, non-parametric Mann-Whitney Rank Sum U-test).

We then asked whether the pattern of synapse-associated protein differed from that in Lkb1-Ret mice. First, fewer Lkb1-Ret cones contained VGLUT1 over time (31.3%, 12.7%, 12.7%, and 3.8% reduction at P1, P3, P5, and P8 respectively, p≤0.01, *Figure 9D*). Second, Lkb1-Ret cones that contained VGLUT1 showed abnormal protein localization. Unlike control animals, VGLUT1 could be found in the soma in addition to the axon (*Figure 9A–C,E*). Indeed, in some cases VGLUT1 was present only in the soma of Lkb1-Ret cones at P5 (15.7% of Lkb1-Ret cones and 0.1% control cones, p<0.0001) even when an axon was present (5.4% of Lkb1-Ret cones and 0.8% control cones, p<0.0001, *Figure 9F–H*). Together, these changes resulted in a 95.2% increase in VGLUT1 mislocalization to the soma of Lkb1-Ret animals relative to controls (p<0.0001, *Figure 9D–F*). Notably abnormal VGLUT1 labeling persisted in Lkb1-Ret animals through P8 even when axon extension defects were largely corrected (p=0.0006, *Figure 6G*, *Figure 9C,F–H*).

To obtain high-resolution views of VGLUT1 localization in control and Lkb1-Ret cones, we again performed expansion microscopy (*Figure 8—figure supplement 1*; *Chen et al., 2015*). We reconstructed VGLUT1 localization within cones in detail and found marked quantitative differences in fluorescent intensity across different neuronal compartments (*Figure 10A*). Relative to controls, VGLUT1 levels were 76.4% and 82.6% decreased in the axon and axon terminal of Lkb1-Ret animals, respectively (p<0.04, *Figure 10B*). In contrast, VGLUT1 levels in the soma of Lkb1-Ret mice exhibited a 62.5% increase in fluorescence (p<0.04, *Figure 10B*). In addition, we noted morphological defects in Lkb1-Ret cones in which VGLUT1 was highly localized to the soma. In these instances, cone soma had a distal bulge in which VGLUT1 was concentrated (*Figure 10C*, associated *Figure 10—videos 1* and *2*). Together, these data suggest that LKB1-induced defects in axon extension impact the localization and distribution of VGLUT1 and other synapse-associated proteins, with decreased localization to the axon and axon terminal and increased localization to the soma.

Based on the synapse protein defects we observed in Lkb1-Ret animals, we questioned whether these ectopic protein patches may correspond with the location of mislocalized neural contacts. To examine this, we used VGLUT1 staining as a readout for synapse protein mislocalization since it was the only protein we identified that was present at high levels during OPL emergence (P3-P5). We first asked whether horizontal cell neurites in Lkb1-Ret animals targeted VGLUT1 presynaptic regions. Horizontal cells and VGLUT1 were co-labeled in control and Lkb1-Ret animals at P5, and the cellular position of VGLUT1 in cones and its apposition to horizontal cell neurites were determined. In control and in Lkb1-Ret mice, the majority of horizontal cell neurites colocalized with VGLUT1 patches (98.5% and 92.7%, respectively; *Figure 11A–B*). Similarly, the majority of VGLUT1 positive regions were contacted by horizontal cell processes (98.5% and 92.9% in control and Lkb1-Ret mice, respectively; *Figure 11C*). We then asked whether the relative location of VGLUT1 patches impacted the fidelity of horizontal cell neurite contacts. Cones, horizontal cells and VGLUT1 were co-labeled in control and Lkb1-Ret animals at P5, and the cellular position of VGLUT1 in cones and its apposition to horizontal cell neurites were determined. In control mice, contacts were largely localized to the terminal (98.4%; *Figure 11A,D*). However, in Lkb1-Ret animals horizontal cell neurites targeted ectopic VGLUT1 presynaptic patches distributed along the cone soma and proximal axon (9.9 fold change in non-terminal contacts; p<0.001; *Figure 11A,D*). These data suggest that synapse-associated protein localization in cones may participate in the restriction of neurite interaction to the cone terminal.

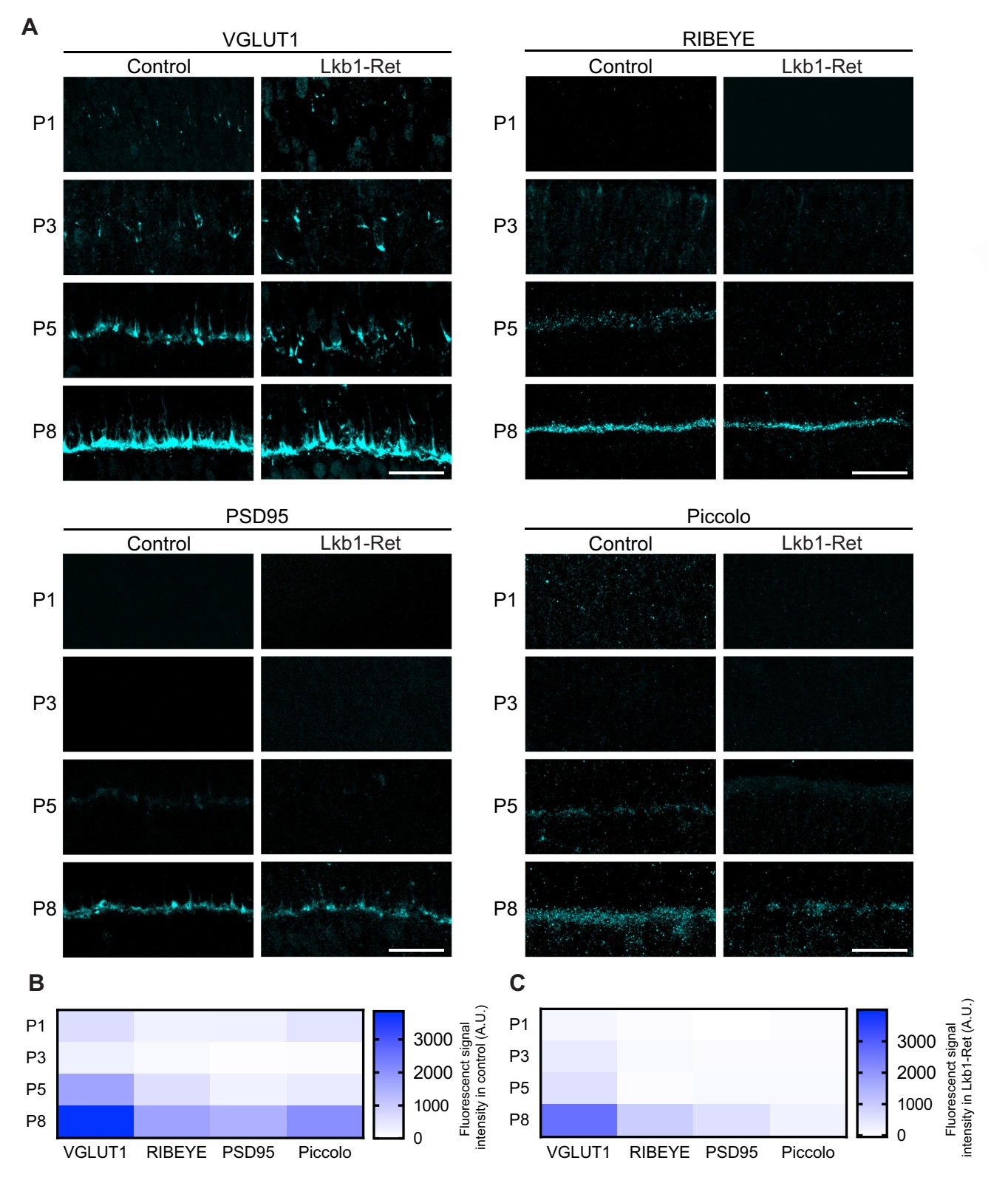

**Figure 7.** Synaptic protein distribution across development. Outer retina synapse-associated proteins were stained and quantified over development to assess levels and localization. (A–C) Representative images (A) and quantification (B–C) of VGLUT1, RIBEYE, PSD95, and Piccolo at P1, P3, P5, and P8 in control and Lkb1-Ret animals. Data in (B) and (C) are presented as a heatmap indicating the corrected total cell fluorescence of each retinal layer

*Figure 7 continued on next page*

*Figure 7 continued*

occupied by the synapse protein signal using a gradient scale where white to blue depicts low to high levels of fluorescent intensity (0–3000, respectively). Scale bars = 25 μm.

## Discussion

The events that accompany synapse formation often overlap in space and time, so it has been difficult to resolve which processes drive the formation of specific patterns of connectivity. Here, we examined the role of LKB1 in synapse layer emergence by utilizing the outer retina where synapses occur at one distinct cellular location. The OPL appears early in development as an ordered cell-free layer that is comprised of contacts between cones and horizontal cells. We show that LKB1 is a key driver of synapse layer emergence independent of AMPK, which is required for adult synapse maintenance. LKB1 deletion results in OPL disorganization and the appearance of small patches interspersed by nuclei that disrupt the synapse lamina. These alterations coincided with specific defects in horizontal cell neurite restriction, which were accompanied by defective axon extension in presynaptic cones. Furthermore, there was a failure of synapse-associated proteins to localize to the axon terminal, and horizontal cell neurites were misdirected to these ectopic synapse protein regions. Together, these data suggest an LKB1-dependent pathway that instructs the timing and location of connectivity in the outer retina via regulation of neuron structure and synapse-associated protein localization.

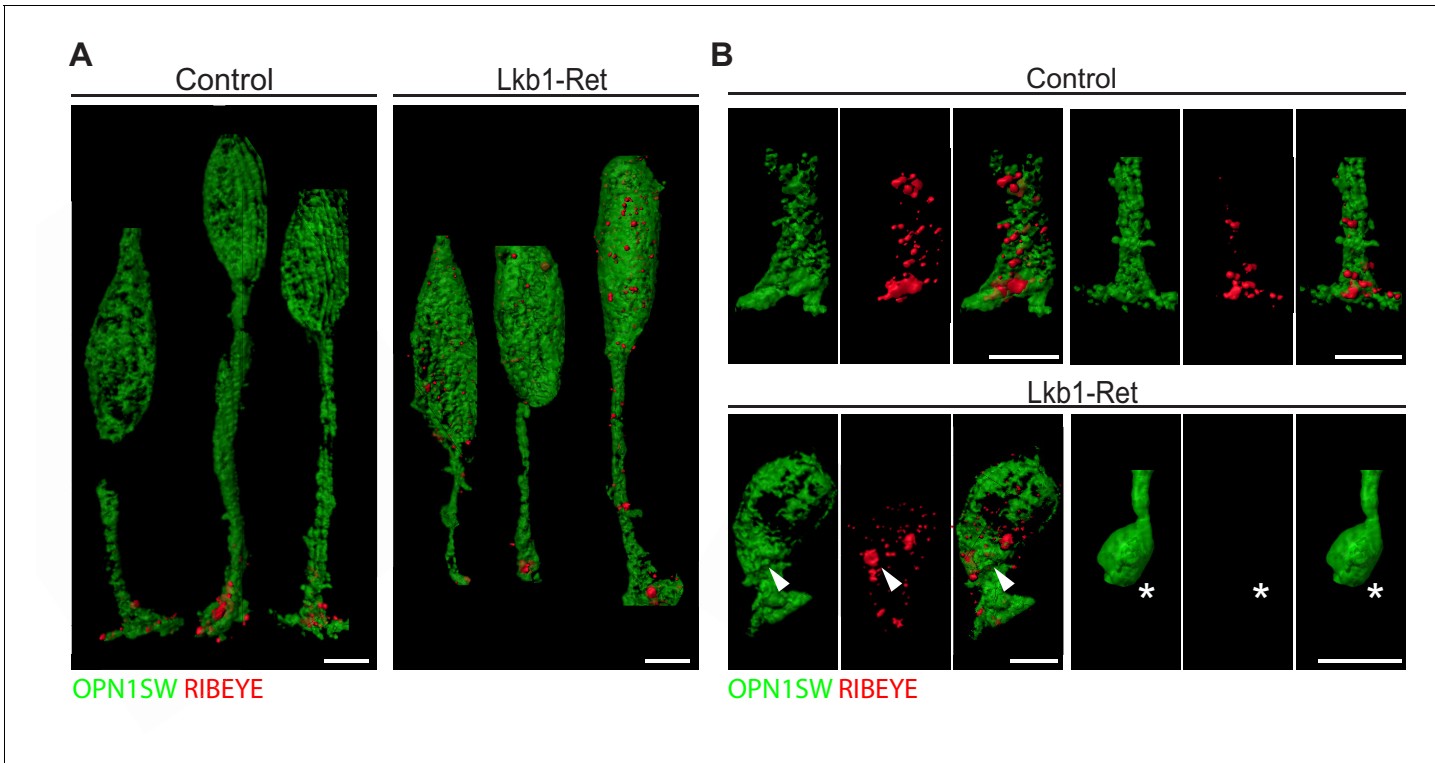

**Figure 8.** Expansion microscopy shows altered RIBEYE localization in LKB1 mutant cones. (**A**) Representative reconstructions of cones from Lkb1-Ret animals and control littermates at P5 are shown following staining with RIBEYE (red) and OPN1SW (green). Samples were expanded, imaged, and reconstructed. (**B**) Reconstructed control and Lkb1-Ret cone terminals show RIBEYE localization defects in Lkb1-Ret animals. Unlike controls, RIBEYE is present in the cell soma (arrowhead) and can be absent from the cone terminal (star). Scale bars = 5 μm.

The online version of this article includes the following figure supplement(s) for figure 8:

**Figure supplement 1.** Expansion microscopy provides increased resolution of retinal neurons.

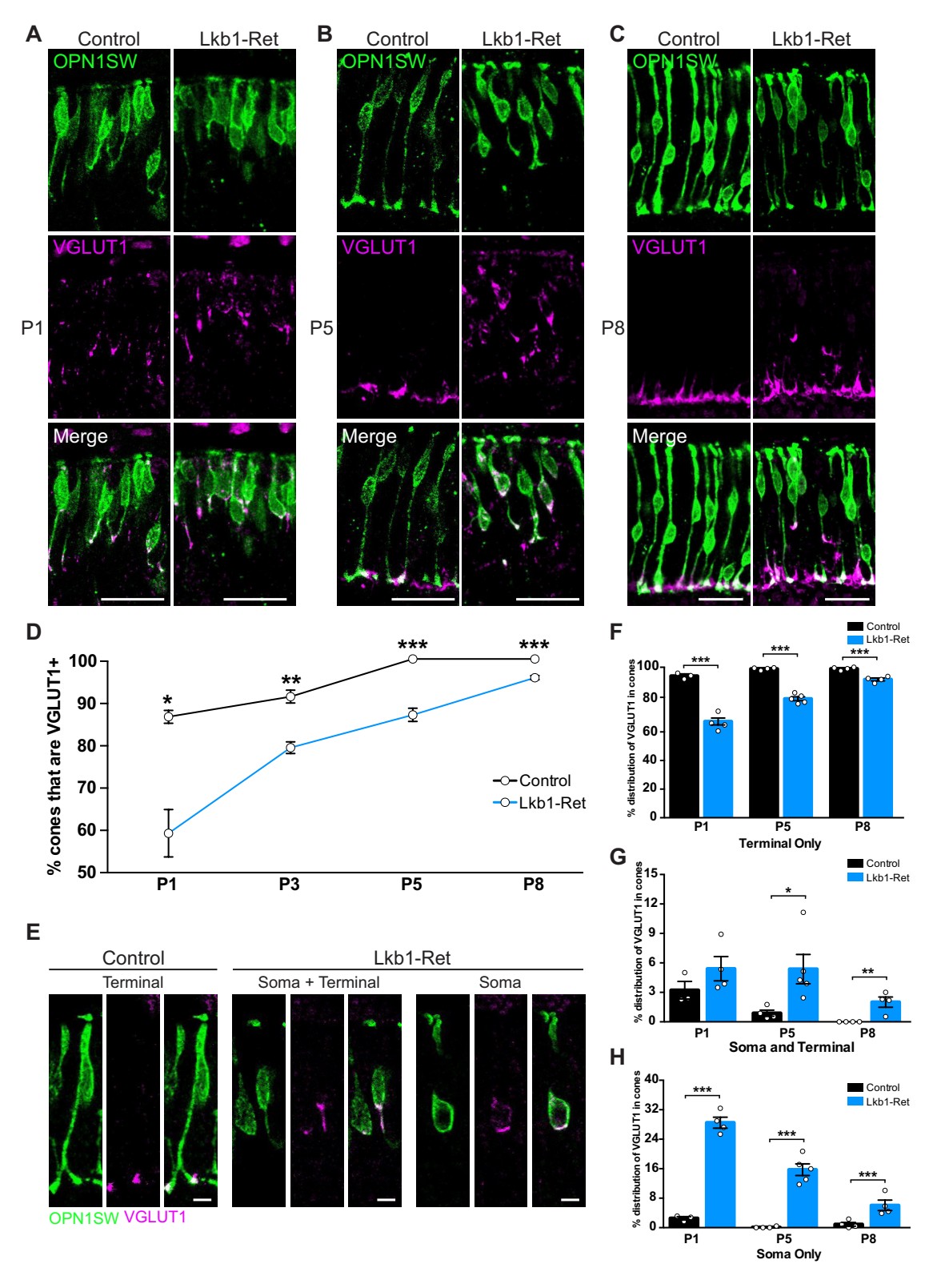

**Figure 9.** LKB1 regulates VGLUT1 levels and localization. Cones were co-labeled with OPN1SW (green) and vesicular glutamate transporter 1 (VGLUT1, magenta) in Lkb1-Ret animals and control littermates. (A–C) At P1 (A), P5 (B), and P8 (C) Lkb1-Ret animals have a decreased number of cone terminals that are VGLUT1 positive and show displaced VGLUT1 localization relative to controls. (D) The number of cones that are VGLUT1 positive is significantly reduced in Lkb1-Ret animals relative to control littermates at all time points. (E) Representative images of VGLUT1 localization in single cones. In

*Figure 9 continued*

control animals, VGLUT1 is found primarily in the terminal (left panel), while Lkb1-Ret animals show cones with abnormal VGLUT1 localization within the cell soma and terminal or within the soma only (right panel). Scale bars = 5 μm. (F–H) Quantification of the number of cones that contain VGLUT1 only within the cone terminal (F), in the terminal and the soma (G), or only within the soma (H) at P1, P5, and P8. N ≥ 3 control and N ≥ 4 Lkb1-Ret animals. Scale bars = 25 μm. Data are represented as the mean ± the s.e.m. ***p<0.001, **p<0.01, *p<0.05, unpaired t-test across rows corrected for multiple comparisons using the Holm-Sidak method.

## LKB1 coordinates neurite remodeling to drive outer retina synapse layer emergence

The morphological steps that characterize OPL development have been well documented (*Sarin et al., 2018*), allowing us to focus our attention on neurons that are first to form nascent synaptic contacts in this region, cones and horizontal cells. Both are born prenatally (*Cepko, 2014*) and then extend neurites that terminate in the future OPL. Our data are consistent with a model in which cones and horizontal cell remodeling regulate the first stages of OPL emergence. We provide several lines of evidence in support of this idea. First, cone axon extension coincides precisely with horizontal cell restriction, and this arrangement is perturbed in LKB1 mutants when axons fail to extend. Second, when horizontal cells eventually refine and cone axons extended in LKB1 mutants, OPL defects were corrected in concert. Other studies also support this model. Zebrafish mutants lacking the synapse protein synaptojanin one show delayed OPL emergence and decreased OPL area that coincided with cone morphological alterations (*Holzhausen et al., 2009*). Notably, these processes are likely distinct from OPL sublamination mechanisms, as this occurs later in development and is regulated by Wnt signaling pathways (*Sarin et al., 2018*). Together, these results suggest that LKB1-driven neurite maturation is important for the precise timing and organization of OPL emergence while other cell types or processes regulate later stages of OPL maturation.

How might cones and horizontal cells interact to regulate synapse layer formation? Horizontal cells undergo marked refinement in coordination with OPL emergence and reorient their organizational axis from a basal/apical orientation to a horizontal one. Several genes have been implicated in horizontal cell birth (*Prox1* and *Foxn4*, *Dyer et al., 2003*; *Li et al., 2004*), migration (*Lhx1*, *Poché et al., 2007*) and neurite confinement in late development and adulthood (*Sema6a* and *PlexinA4*, *Matsuoka et al., 2012*, *Cacna1*, *Dick et al., 2003*, and *Bassoon*, *Bayley and Morgans, 2007*). However, the mechanisms that are responsible for horizontal cell reorientation and refinement are unknown. Our data suggest a model in which LKB1-driven cone axon extension and horizontal cell arbor refinement are coordinately regulated. In support of this idea, mislaminated horizontal cell processes maintained contact with mistargeted cone axons in LKB1 mutants, and horizontal cell lamination defects were corrected in precise coordination with cone axon extension. Further, deletion of LKB1 in the retina generally, but not in horizontal cells specifically, led to horizontal cell refinement defects. Together, these data suggest that refinement of neurites from a basal/apical orientation to a horizontal orientation early in development may depend on LKB1-mediated contact with developing cone axons.

## Defective synapse-associated protein localization in LKB1 mutant mice

How might presynaptic neurite growth regulate synapse layer formation in the outer retina? Our studies suggest terminal maturation and synapse-associated protein levels or localization may participate. In particular, VGLUT1 showed early and persistent localization defects. VGLUT1 is a vesicular glutamate transporter, and cones rely on the vesicular release of glutamate for synaptic neurotransmission after axon development is complete (*Sherry et al., 2003*; *Fremeau et al., 2004*; *Andreae and Burrone, 2014*). Consistent with this, VGLUT1 is required for proper glutamate transmission and photoreceptor activity (*Johnson et al., 2007*; *Wojcik et al., 2004*). VGLUT1 levels have also been tied to synapse emergence in the cortex downstream of MECP2 (*Chao et al., 2007*). Other studies have also found links between synapse protein localization, axon extension, and synapse formation. For example, neurotransmitter vesicle fusion and release have been shown to be important for axon outgrowth (*Feng et al., 2002*). Interestingly, the neuromuscular junction requires L-type calcium channels to regulate synapse formation. Loss of these calcium channels results in loss of acetylcholine receptor patterning, as well as excessive nerve branching and failure of axons to

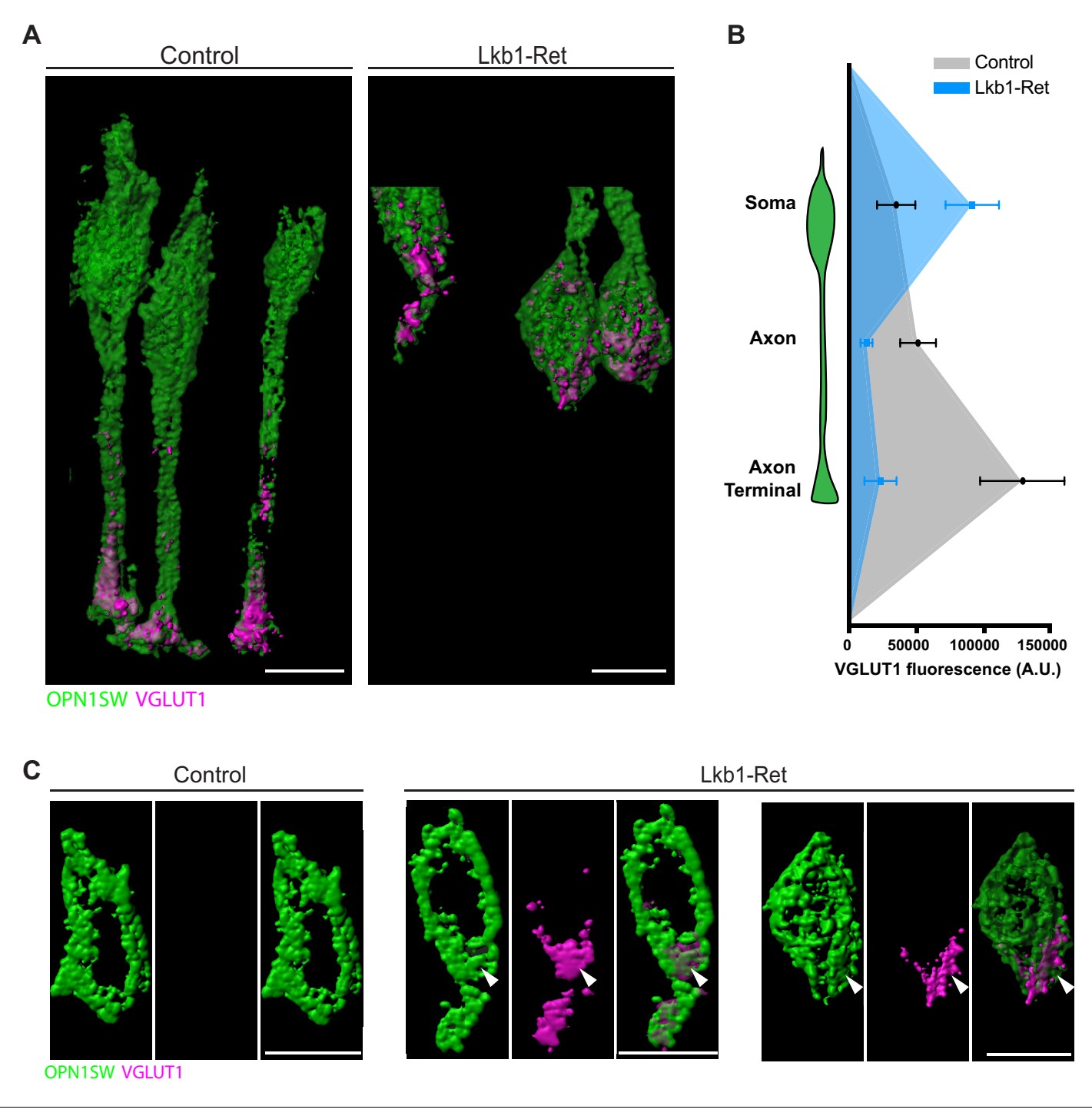

**Figure 10.** VGLUT1 is mislocalized in LKB1 mutant cone somas. (**A**) Representative reconstructions of cones from Lkb1-Ret animals and control littermates at P5 are shown following staining with VGLUT1 (magenta) and OPN1SW (green). Samples were expanded, imaged, and reconstructed. (**B**) Quantification of VGLUT1 expression in the soma, axon, and axon terminal of cones in both control and Lkb1-Ret animals. Data are presented as a line graph indicating the corrected total cell fluorescence within each neuron compartment occupied by VGLUT1 signal. (**C**) Reconstructed control and Lkb1-Ret cone somas show morphological defects in Lkb1-Ret animals. VGLUT1 is highly localized to a bulge associated with the cell soma (arrowhead). Scale bars = 5 μm. Data are represented as the mean ± the s.e.m.

The online version of this article includes the following video(s) for figure 10:

**Figure 10—video 1.** Reconstructed expanded cones reveal VGLUT1 distribution in control mice.

https://elifesciences.org/articles/56931#fig10video1

*Figure 10 continued on next page*

*Figure 10 continued*
**Figure 10—video 2.** Reconstructed cones reveal mislocalized VGLUT1 in LKB1 mutant cone soma.
https://elifesciences.org/articles/56931#fig10video2

recognize postsynaptic targets (*Kaplan et al., 2018*; *Kaplan and Flucher, 2019*). It is also possible that LKB1 may directly regulate synapse protein localization independent of cone axon extension, as VGLUT1 and RIBEYE were mislocalized in cones even when axons were present. Consistent with this idea, LKB1 has been shown to be involved in the polarized transport of proteins in *Drosophila* epithelial cells (*Jang et al., 2008*) and mouse hepatocytes (*Homolya et al., 2014*). Together, these data suggest that axon extension, terminal maturation, and polarized synapse protein transport might be molecularly coordinated and implicate LKB1 as a regulator of these events.

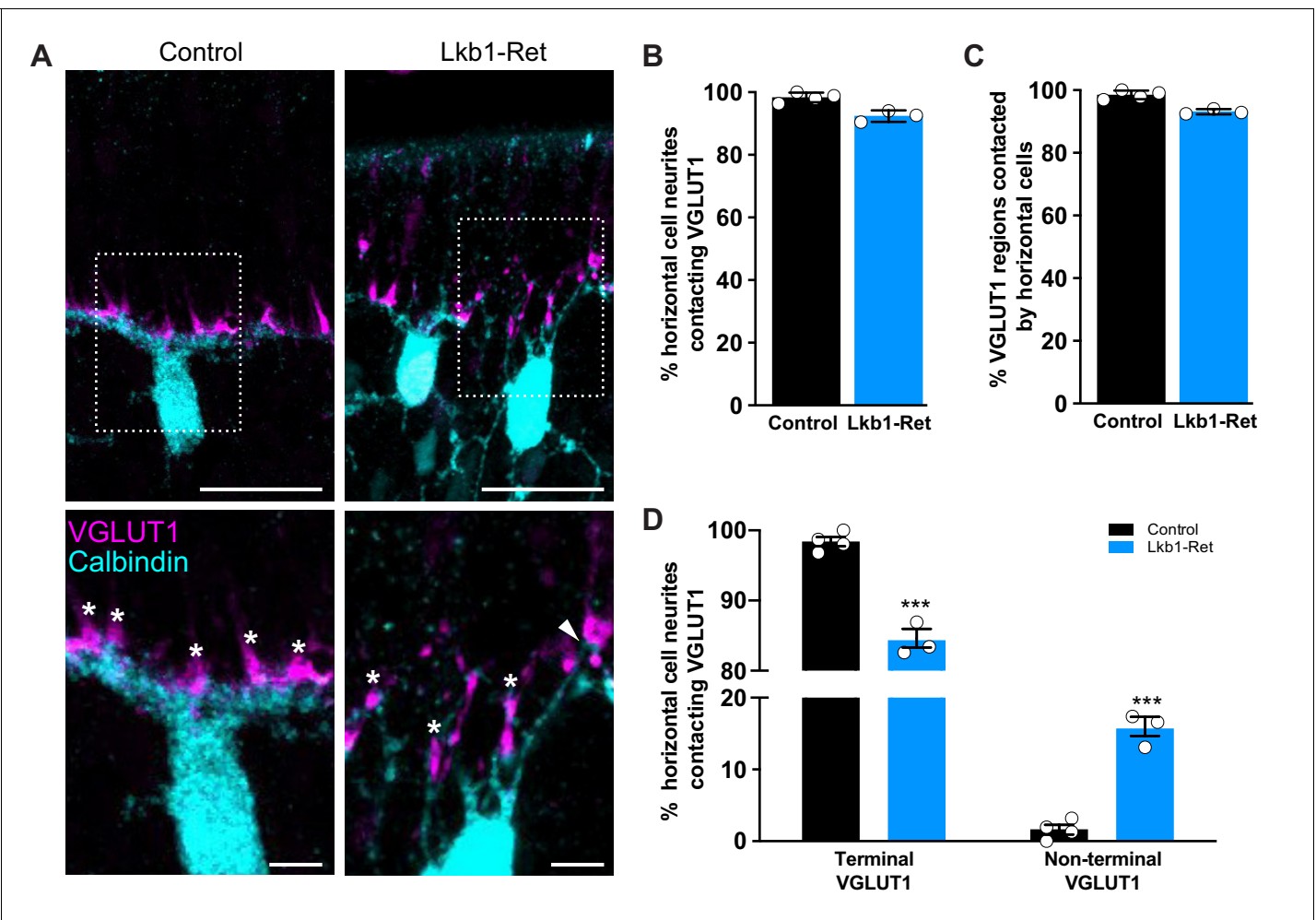

**Figure 11.** Horizontal cell neurites contact ectopic VGLUT1 locations. (A) Representative images of horizontal cells (calbindin, cyan) and VGLUT1 (magenta) contacts at P5 in Lkb1-Ret mice and littermate controls are shown. The boxed area is presented as a higher magnification image highlighting horizontal cells contacting VGLUT1 at cone terminals (star) and at non-terminal cone positions (arrow). (B–C) The percent of horizontal cell processes contacting VGLUT1 (B) and the percent of VGLUT1 regions that were contacted by horizontal cells (C) were quantified at P5. Nearly all horizontal cell processes contacted VGLUT1 and nearly all VGLUT1 positive regions were contacted by horizontal cell processes. The relative percent of contacts did not significantly differ between control and Lkb1-Ret animals. N = 4 control and N = 3 Lkb1-Ret animals. (D) The relative location of horizontal cell processes contacting VGLUT1 at cone terminals and non-terminal cone positions were quantified at P5. There is a significant increase in horizontal cell processes contacting non-terminal VGLUT1 in cones in Lkb1-Ret animals compared to control. N = 4 control and N = 3 Lkb1-Ret animals. Scale bars = 25 μm (upper panel) and 5 μm (lower panel). Data are represented as the mean ± the s.e.m. (B,C, non-parametric Mann-Whitney Rank Sum U-test; D, ***p<0.001, unpaired t-test across rows corrected for multiple comparisons using the Holm-Sidak method).

## Diverse roles for LKB1

LKB1 is a serine/threonine kinase that is best known for its tumor suppressor functions (*Shackelford and Shaw, 2009*), but it is increasingly recognized as a key regulator of the nervous system (*Kuwako and Okano, 2018*). Though it is expressed broadly throughout the CNS, its function appears remarkably dependent upon both neuron type and maturation stage. Perhaps its best studied role is in axon development, but even here, mechanisms differ. In hippocampal and cortical neurons LKB1 deletion impairs axon initiation (*Barnes et al., 2007*; *Shelly et al., 2007*) and branching (*Courchet et al., 2013*). In contrast, LKB1 is dispensable for axon formation in the brainstem and spinal cord, which depend instead on SAD kinases (*Lilley et al., 2013*). In developing outer retina, LKB1 had no detectable effect on outer retina neuron cell numbers but was required for cone axon initiation and elongation as well as OPL emergence. Notably, both in retina and in brain, LKB1-deficient neurons do eventually grow axons that approximate the lengths reached in controls, suggesting that LKB1-independent pathways participate in axon growth in diverse neuron types. Thus, LKB1 is required at different stages of axon development that may point to distinct LKB1-dependent and independent mechanisms by which neuron subtypes generate and maintain polarity.

Our results suggest at least two additional functions for LKB1 in neurons: synapse layer formation and synapse protein localization. What mechanisms might be at play? Unlike rod aging, which requires AMPK downstream of LKB1 (*Samuel et al., 2014*), cone-axon extension appears independent of AMPK signaling. LKB1 may instead impact developmental processes through regulation of microtubule formation or motor protein movement via its role in MARK activation (*Nakano et al., 2010*; *Biernat et al., 2002*). Alternatively, LKB1 also regulates energy production in response to mechanical force (*Bays et al., 2017*), which may be important for physically anchoring pre and post-synaptic terminals together. In future studies it will be interesting to determine the distinct, neuron-specific downstream pathways that LKB1 engages.

In summary, we have uncovered a key molecular regulator responsible for the emergence of ordered connectivity in the outer retina. We have also shown how the localization of pre and post-synaptic arbors can sculpt the emergence of synapse lamina and shed light on the role of neurite remodeling and synapse-associated protein localization in these events. Particularly fascinating is the apparent cellular specificity with which LKB1 functions despite its ubiquitous expression (*Samuel et al., 2014*). This suggests additional levels of regulation that could be inherent to the diversity in neuron developmental processes, the localization of LKB1 itself, or the presence or absence of downstream signaling molecules.

## Materials and methods

### Key resources table

| Reagent type (species) or resource | Designation | Source or reference | Identifiers | Additional information |
|---|---|---|---|---|
| Gene (*Mus musculus*) | *Stk11* | Mouse Genome Informatics | MGI:1341870 NCBI Gene: 20869 | |
| Gene (*Mus musculus*) | *Prkaa1* | Mouse Genome Informatics | MGI:2145955 NCBI Gene: 105787 | |
| Gene (*Mus musculus*) | *Prkaa2* | Mouse Genome Informatics | MGI:1336173 NCBI Gene: 108079 | |
| Strain, strain background (*Mus musculus*, male and female) | *Stk11<sup>F/F</sup>* (previously called *Lkb1<sup>F/F</sup>*) | *Bardeesy et al., 2002*; DOI: 10.1038/nature01045 | | |
| Strain, strain background (*Mus musculus*, male and female) | *Prkaa1 <sup>F/F</sup> Prkaa2 <sup>F/F</sup>* | *Nakada et al., 2010*; DOI: 10.1038/nature09571 | | |
| Strain, strain background (*Mus musculus*, male and female) | *Vsx2-Cre* (previously called *Chx10-Cre*) | The Jackson Laboratory | RRID:IMSR_JAX:005105 | |

*Continued on next page*

*Continued*

| Reagent type (species) or resource | Designation | Source or reference | Identifiers | Additional information |
|---|---|---|---|---|
| Strain, strain background (*Mus musculus*, male and female) | *Gja10-ires-iCre* (previously called *Cx57-ires-iCre*) | *Hirano et al., 2016*; DOI: 10.1523/ENEURO.0148–15.2016 | | |
| Antibody | Chicken polyclonal anti-Calbindin | Novus Biologicals | NBP2-50028 | IHC (1:2000) |
| Antibody | Rabbit polyclonal anti-Calbindin | Swant | CB38; RRID:AB-100000340 | IHC (1:10000) |
| Antibody | Sheep polyclonal anti-Chx10 | Exalpha | X1180P; RRID:AB_2314191 | IHC (1:300) |
| Antibody | Goat polyclonal anti-OPN1SW | Santa Crus | Sc-14363; RRID:AB_2158332 | IHC (1:500) |
| Antibody | Rabbit polyclonal anti-Piccolo | Synaptic Systems | 142003; RRID:AB_2160182 | IHC (1:500) |
| Antibody | Mouse monoclonal anti-Protein Kinase C α | Abcam | Ab31; RRID:AB_303507 | IHC (1:500) |
| Antibody | Goat polyclonal anti-PSD95 | Abcam | Ab12093; RRID:AB_298846 | IHC (1:500) |
| Antibody | Mouse monoclonal anti-rhodopsin | Abcam | Ab98887; RRID:AB_10696805 | IHC (1:500) |
| Antibody | Rabbit polyclonal anti-RIBEYE | Synaptic Systems | 192103; RRID:AB_2086775 | IHC (1:500) |
| Antibody | Rabbit polyclonal anti-VGLUT1 | Synaptic Systems | 135302; RRID:AB_887877 | IHC (1:500) |
| Commercial assay or kit | RNeasy Mini Kit | Qiagen | Cat. No. 74104 | |
| Commercial assay or kit | RNAscope Multiplex Fluorescent v2 | ACDbio | Cat. No. 323120 | |
| Other | RNAscope Probe-Mm-Stk11 | ACDbio | Cat. No. 469211 | |
| Chemical compound, drug | Acryloyl-X, SE | Thermofisher | Cat. No. A20770 | |
| Chemical compound, drug | TEMED | Thermofisher | Cat. No. 17919 | |
| Chemical compound, drug | APS | Sigma Aldrich | Cat. No. 248614 | |
| Chemical compound, drug | 4-Hydroxy-TEMPO | Sigma Aldrich | Cat. No. 176141 | |
| Chemical compound, drug | Sodium Acrylate | Sigma Aldrich | Cat. No. 408220 | |
| Chemical compound, drug | Acrylamide | Sigma Aldrich | Cat. No. A9099 | |
| Chemical compound, drug | N-N'-Methyl enebisacrylamide | Sigma Aldrich | Cat. No. M7279 | |
| Chemical compound, drug | Sodium Chloride | Sigma Aldrich | Cat. No. S9888 | |
| Software, algorithm | ImageJ | NIH | https://imagej.nih.gov/ij/; RRID:SCR_003070 | |
| Software, algorithm | Imaris 7 | Oxford Instruments | RRID:SCR_007370 | |
| Software, algorithm | Prism7 | Graphpad | http://www.graphpad.com; RRID:SCR_002798 | |

## Mouse strains

Mouse strain names were modified as per eLife's gene nomenclature policy. The *Stk11* (*Lkb1*) conditional null mutant *Stk11$^{F/F}$* has been described previously (*Bardeesy et al., 2002*) and was provided by R. DePinho, MD Anderson Cancer Center. In this strain, *loxP* sequences flank exons 2–6, resulting in a complete loss of LKB1 function. To broadly delete LKB1 in the retina, *Stk11$^{F/F}$* mice were crossed to *Vsx2-Cre* (*Rowan and Cepko, 2004*), provided by C. Cepko, Harvard University) to generate animals referred to here as Lkb1-Ret mice. To delete LKB1 in horizontal cells we crossed *Stk11$^{F/F}$* animals to the *Gja10-ires-iCre* line (*Hirano et al., 2016*) to generate animals referred to here as Lkb1-HC mice. For these lines, *Stk11$^{F/F}$* littermates were used as controls. To look at the role of AMPK, we used the conditional null mutant *Prkaa1$^{F/F}$Prkaa2$^{F/F}$* which has been previously described (*Nakada et al., 2010*) and was provided by Dr. Daisuke Nakada, Baylor College of Medicine. To broadly delete both alpha subunits of *Ampk* (*Prkaa1 and Prkaa2*) in the retina, *Prkaa1$^{F/F}$Prkaa2$^{F/F}$* mice were crossed to *Vsx2-Cre* to generate animals referred to here as Ampk-Ret. For these lines, *Prkaa1$^{F/F}$Prkaa2$^{F/F}$* littermates were used as controls. Experiments were carried out in male and female mice in accordance with the recommendations in the Guide for the Care and Use of Laboratory Animals of the NIH under protocols approved by the BCM Institutional Animal Care and Use Committee.

## Immunohistochemistry

Eyes were collected from animals at P1, P3, P5, P8, and P14. The day of birth was designated as postnatal day 0 (P0). Whole eyes were fixed for 45 min in 4% paraformaldehyde and then rinsed with PBS. Retina cross sections were prepared as described (*Samuel et al., 2014*). Briefly, eye cups were dissected, and the cornea and lens were removed. The samples were then cryoprotected in 30% sucrose, embedded in Optimal Cutting Temperature (OCT) compound (Sakura, Torrance, CA), frozen in methyl butane on dry ice, sectioned at 20 µm, and then mounted on Superfrost Plus slides (VWR). Slides were incubated with blocking solution (3% normal donkey serum and 0.3% Triton X-100 in PBS) for 1 hr, and then with primary antibodies (*Table 1*) O/N at 4°C. Slides were washed with PBS three times for 10 min and incubated with secondary antibodies (Jackson ImmunoResearch Laboratories, West Grove, PA) for 1 hr at room temperature. Slides were then washed with PBS three times for 10 min. All samples were mounted in Vectashield (Vector Laboratories, Burlingame, CA). The images were acquired on an Olympus Fluoview FV1200 confocal microscope and processed using Fiji. For cone and horizontal cell neuron reconstruction, 3D rendered images of cones were generated using Imaris.

## Expansion microscopy

Expansion microscopy was performed as previously described (*Asano et al., 2018*). In brief, tissue samples were prepared for immunohistochemistry as described above. After the final PBS wash, samples were fully immersed in a 0.1 mg/mL solution of Acryloyl-X, SE (Thermofisher #A20770) for four hours at room temperature. Following this, samples were washed three times in 1X PBS for 10 min prior to proceeding with gelation to remove unreacted reagents. To form the hydrophilic gel around the samples, a 47:1:1:1 solution of Stock X (see below): TEMED (0.1 mg/mL, Thermo Fisher #17919): APS (0.1 mg/mL, Sigma Aldrich #248614–5G): 4-Hydroxy-TEMPO (5.0 mg/mL, Sigma Aldrich #176141–1G) was placed within chambers surrounding the tissue slices. Stock X solution was made using the following proportions: 38 grams of sodium acrylate (Sigma Aldrich #408220–5G), 50 grams of acrylamide (Sigma Aldrich #A9099-25G), 2 grams of N,N'-Methylenebisacrylamide (Sigma Aldrich #M7279-25G), and 29.2 grams of sodium chloride (S9888-25G) in a total volume of 100 mL using 10X PBS stock. Slides were then incubated at 4°C for one hour followed by incubation at 37°C for an additional three hours to allow the gel to fully set. Once solidified, slides were trimmed of excess gel and placed into deionized water for one hour. To allow the samples to fully expand, the water bath was exchanged every fifteen minutes with fresh deionized water. Individual gels were then mounted in deep chambers with fresh deionized water surrounding the sample and imaged on an Olympus Fluoview FV1200 confocal microscope and processed using Fiji. For neuron reconstruction, 3D rendered images were generated using Imaris.

## In situ hybridization

In situ hybridization was performed by the RNA In Situ Hybridization Core at BCM using an automated robotic platform as previously described (*Yaylaoglu et al., 2005*). In brief, we prepared digoxigenin (DIG)-labeled riboprobes using cDNA from E15 and P7 mouse brain RNA using a RNeasy Mini Kit (Qiagen, Germany). DNase treatment was performed to digest genomic DNA. The following LKB1 forward and reverse PCR primers were used to generate cDNA fragments corresponding to the desired ribo-probes: GCGATTTAGGTGACACTATAGCTTTTCAGGTTTCAAGG TGGAC and GCGTAATACGACTCACTATAGGGACCCTCATAGCCATAGCTCAAA. DIG-labeled riboprobes were synthesized using a DIG RNA labeling kit (Roche, Switzerland), diluted in hybridization buffer at a concentration of 100 ng/uL, and stored at −20℃ until use.

Eyes were enucleated, and the lens was dissected out. Eyecups were cryoprotected in 30% sucrose, frozen in OCT (VWR), and stored at −80℃ before sectioning. Retina cryosections (20 μm) were mounted on Superfrost Plus Slides (VWR). Sections were fixed and acetylated before the hybridization procedure, which was performed on a high-throughput platform. The slides were developed using tyramide labeled with Cy3 directly (TSA-Plus system; Perkin-Elmer Life Sciences, Waltham, MA) for 15 min, followed by staining with 4'−6-diamindino-2-phenlindole (DAPI) before mounting in Prolong Diamond (Invitrogen, Carlsbad, CA).

## RNAScope

To confirm deletion of *Stk11*, RNAScope was performed using RNAscope Probe-Mm-Stk11 (cat. # 469211) on 20 μm tissue sections collected as described above for immunohistochemistry. After sectioning, 4% paraformaldehyde was applied to each slide for an additional 30 min at room temperature. Fluorescent in situ hybridization (FISH) was performed using the commercially available RNAscope fluorescent multiplex assay according to the manufacturer's instructions (ACD-bio, Newark, CA) with the following with minor modifications. Tissue was dehydrated using an ethanol gradient of 10%, 30%, 50%, 70% and 100% (3 minutes each), and the boiling time in target retrieval solution was shortened to 5 min. After FISH, slides were co-stained for calbindin to visualize horizontal cell bodies.

## Histological quantification

All quantification was performed using retinal sections prepared from Lkb1-Ret, Lkb1-HC, and control animals at early postnatal ages (P1, P3, P5, P8). Littermate controls were used in all experiments, and all images were acquired at equivalent retinal eccentricities from the optic nerve head. For all experiments, data were collected from 3 to 8 mice per group, and three to four images per animal were obtained. To quantify the number of ectopic nuclei in the OPL, DAPI was used to label nuclei at P8, and antibodies to rhodopsin (rods) and Chx10 (bipolar cells) were used to demarcate the OPL boundaries. The number of nuclei within the OPL was counted in each image (211.97 × 211.97 μm$^2$), and values were averaged. To quantify the number of OPL patches relative to cone terminals and horizontal cell contacts, nuclei were stained with DAPI while cones were stained with anti-OPN1SW. Patches were defined as the location of a cone terminal that coincided with a gap in the nuclear plexus. The distance of both patches and cone terminals were quantified from the apical retina surface as the shortest line that encompassed the apical surface and the top of the OPL patch. To quantify fluorescent intensity for in situ hybridization and synapse-associated proteins, retinal, layer boundaries were manually defined using the corresponding DAPI images. For P5, P8, and P14 retinas, the boundaries of the ONL, OPL, INL, IPL, and GCL were marked. For P1-P3, the ONBL was divided into equal sublayers to allow for more spatial definition of the expression pattern. The relative levels of the signal averaged over three randomly selected regions of the same size within each layer were computed, and a background subtraction was applied to remove background noise. To quantify the number of bipolar cells, cones, and horizontal cells, antibodies against Chx10, OPN1SW, and calbindin were used, respectively. The number of cells within an image (211.97 × 211.97 μm$^2$) were counted and values were averaged. To quantify the number of apical neurites per horizontal cell, horizontal cells were stained with an antibody to calbindin. The number of neurites coming from the top half of each horizontal cell soma was counted, and values for each horizontal cell in each image were averaged. To quantify cone axon length, sections were stained with an OPN1SW antibody to label cone photoreceptors. The length of each axon terminal was defined as

the shortest line that encompassed the base of the nuclei and the end of the axon terminal. All cones in a z-stack were analyzed. To quantify the levels and location of VGLUT1, sections were stained with an antibody to VGLUT1 and colabeled with OPN1SW to mark cone cell bodies and terminals. A cone with VGLUT1 staining present within the cell body was quantified as a soma VGLUT1 containing cell, while a cell with VGLUT1 staining that overlapped with the terminal was quantified as a terminal VGLUT1 containing cell. To quantify the corrected total cell fluorescence of VGLUT1 in different neuronal compartments, the area of the cell multiplied by the mean fluorescence of background readings was subtracted from the integrated density. The number of horizontal cell contacts made with cones was determined in a single optical section, and the distance of this contact was measured from the end of the horizontal cell neurite to the center of the horizontal cell soma. Contacts with terminals were denoted by horizontal cell neurites overlapping with cone terminals, while contacts with non-terminal regions were denoted by horizontal cell neurites overlapping with either the cone soma, axon shaft, or inner and outer segments. Horizontal cell colocalization with VGLUT1 was determined by examining the overlap between calbindin and VGLUT1. Horizontal contacts with VGLUT1 at neuron terminals were denoted by calbindin positive neurites overlapping with terminal restricted VGLUT1, while contacts with non-terminal regions were denoted by horizontal cell neurites overlapping with VGLUT1 localized in either the cone soma or along the axon shaft. Analysis of RNAscope was conducted by counting *Stk11* positive puncta within horizontal cell bodies across sections where samples were blinded for identity.

## Quantification algorithms

The area of the OPL and horizontal cell terminal restriction were measured using algorithms developed for this purpose in Fiji. All algorithms were validated by manual quantification of representative data sets prior to use. OPL area was measured by selecting the DAPI channel and normalizing the saturation using the normalize equalize function set at 0.3 to remove noise in individual cells. Gaussian blur was then applied with sigma = 5. The image was then converted to binary and inverted to count the empty OPL space using the analyze particles function. For each animal, images of 211.97 $\times$ 211.97 $\mu m^2$ from 2 to 4 different locations were included, and for each image 3 individual Z slices at least 3 $\mu m$ apart were analyzed. All data were then averaged across each sample and converted from pixels to microns before statistical analysis. Horizontal cell terminal restriction was measured in Z stacks that were condensed using the maximum intensity function. Outliers within one radius and over a brightness threshold of 50 were removed. The image was then rotated 90 degrees, and the intensity across each region was summed in bins following alignment to the highest intensity as the center of the restriction. For each animal, images from 2 to 4 different locations were analyzed.

## Statistical analysis

Analyses of the number of ectopic nuclei, number of outer retina neurons, OPL area, number of apical neurites per horizontal cells, total number of patches, axon length, number of contacts between horizontal cells and cones or VGLUT1, number of *Stk11* puncta per horizontal cell, and the length of horizontal cell neurites that contact cones were performed using a non-parametric Mann-Whitney Rank Sum U-test. Analyses of the distance of patches from apical surface were performed using an unpaired two-tailed Student's t test. Horizontal cell restriction, VGLUT1 localization and expression, and the position of horizontal cell and cone or VGLUT1 contacts were analyzed using an unpaired t-test across rows, with VGLUT1 quantifications and position of contacts corrected for multiple comparisons using the Holm-Sidak method. Statistical differences were evaluated using GraphPad Prism seven software. $p < 0.05$ was considered statistically significant.

## Acknowledgements

We thank Jeannie Chin, Matthew Rasband, Ross Poché, Kartik Venkatachalam, Elizabeth Zuniga-Sanchez, and members of our laboratory for scientific discussions and advice. We thank Hui Zheng's laboratory for the use of Imaris. This work was supported by the National Institutes of Health (NIH, R00AG044444, DP2EY02798, 1R56AG061808-01, and R01 EY030458-01 to MAS), the Cancer Prevention Research Institute of Texas, and the Brain Research Foundation. CAB was supported by the NIH and the National Eye Institute under award number T32EY007001. NEA was supported by the NIH and the National Institute of General Medical Sciences under award number T32GM088129.

This project was also supported by the RNA In Situ Hybridization Core facility at BMC with the expert assistance of Cecilia Ljungberg, Ph.D., and funding from the NIH (1S10 OD016167, IDDRC grant 1U54 HD083092, and the Eunice Kennedy Shriver National Institute of Child Health and Human Development).

## Additional information

### Funding

| Funder | Grant reference number | Author |
| --- | --- | --- |
| National Institute on Aging | 1R56AG061808-01 | Melanie A Samuel |
| National Eye Institute | R01 EY030458-01 | Melanie A Samuel |
| Ted Nash Long Life Foundation | | Melanie A Samuel |
| Brain Research Foundation | | Melanie A Samuel |
| National Eye Institute | DP2EY027984-02 | Melanie A Samuel |
| National Eye Institute | T32EY007001 | Courtney A Burger |
| National Institute of General Medical Sciences | T32GM088129 | Nicholas E Albrecht |
| National Institute on Aging | R00AG044444 | Melanie A Samuel |
| Cancer Prevention Research Institute of Texas | | Melanie A Samuel |

The funders had no role in study design, data collection and interpretation, or the decision to submit the work for publication.

### Author contributions

Courtney A Burger, Conceptualization, Data curation, Formal analysis, Validation, Investigation, Visualization, Methodology, Writing - original draft, Project administration, Writing - review and editing; Jonathan Alevy, Justine H Liang, Investigation, Visualization; Anna K Casasent, Data curation, Software, Formal analysis; Danye Jiang, Nicholas E Albrecht, Data curation, Formal analysis, Investigation; Arlene A Hirano, Nicholas C Brecha, Resources, Writing - review and editing; Melanie A Samuel, Conceptualization, Resources, Data curation, Supervision, Funding acquisition, Methodology, Writing - original draft, Project administration, Writing - review and editing

### Author ORCIDs

Nicholas E Albrecht  https://orcid.org/0000-0001-8045-2290
Arlene A Hirano  http://orcid.org/0000-0001-8842-3582
Melanie A Samuel  https://orcid.org/0000-0002-4804-2491

### Ethics

Animal experimentation: Experiments were carried out in male and female mice in accordance with the recommendations in the Guide for the Care and Use of Laboratory Animals of the NIH under protocols approved by the BCM Institutional Animal Care and Use Committee (AN6785). Every effort was made to minimize animal suffering.

### Decision letter and Author response

Decision letter https://doi.org/10.7554/eLife.56931.sa1
Author response https://doi.org/10.7554/eLife.56931.sa2

# Additional files

## Supplementary files

- Source code 1. OPL area quantification script.

- Source code 2. Neurite quantificaiton script.

- Transparent reporting form

## Data availability

Source data analysis code have been provided from Figures 1-4.

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
