## [Decision Letter]

**Acceptance summary:**

The manuscript by Burger et al. explores the role of the serine/threonine kinase LKB1 in neuronal remodeling during the development of the mouse retina. The experiments here are well designed and executed, and the manuscript is clearly written. The take-home lesson is that defects in cone extension, caused by an absence of LKB1 in cones in the LKB1RET knockout mice and most obvious at P1-P5, cause defects in synapse localization and HC remodeling. This suggests that in the WT animals, normal cone extension drives HC neurite remodeling.

**Decision letter after peer review:**

[Editors’ note: the authors submitted for reconsideration following the decision after peer review. What follows is the decision letter after the first round of review.]

Thank you for submitting your work entitled "LKB1 coordinates neurite remodeling to drive synapse layer emergence in the outer retina" for consideration by *eLife*. Your article has been reviewed by three peer reviewers, and the evaluation has been overseen by a Senior Editor. The reviewers have opted to remain anonymous. Our decision has been reached after consultation between the reviewers. Based on these discussions and the individual reviews below, we regret to inform you that your work will not be considered further for publication in *eLife*.

Summary:

As you will see, the reviewers were very interested in role of LKB1 in neurite remodeling in the retina, and the generally high quality of your work. One of the reviewers was generally satisfied with the manuscript. However, the other two reviewers had concerns that focus around the impact of this work compared to your prior paper in 2014; a number of specific technical questions; and concerns with the evidence to support some of your conclusions. In particular, they thought the role of VGlut1 was overstated and would require substantially more evidence, and they were unconvinced by the experiments showing that LKB1 acts non-cell autonomously in horizontal cells. The full original comments of the reviewers are below.

Reviewer #1:

A very nice study exploring the cellular and molecular mechanisms of synapse and layer formation in the outer mammalian retina. This is an excellent model for parsing these processes more generally. Overall the study appears carefully done and clearly shows a role for LKB1 in horizontal cell neurite restriction and the timing of layer formation.

I have 2 questions that I hope the authors can address in the Discussion or Results.

1) Do they think (or have evidence for the idea) that horizontal cells are targeting VGLUT1 per se, or their terminals?

2) Is there anything about the sequence or structure of VGLUT1 that makes this a real possibility? Similar mechanisms have, after all, been observed for L-Type Ca^2+^ channels at NMJs.

Reviewer #2:

This manuscript by Burger et al. explores the role of the serine/threonine kinase LKB1 in the development of the mouse retina. LKB1 has been previously shown to be important in a number of aspects of neuronal development generally, and has been shown to be widely expressed in retinal cells in aging animals, but LKB1's role in the refinement of retinal cell architecture and synapse localization is a novel finding of this manuscript. Overall, the work is generally interesting to the field, though its impact could be improved with some additional experiments.

1) I am perplexed/alarmed by the many reported p-values of p=0.0286. Specifically Figure 1C is reported as showing a 40.1% reduction in number of patches at P3, p = 0.0286; Figure 2A, B is reported as showing an 85.6% increase in the number of ectopic nuclei at P8, p = 0.0286; Figure 2C, D is reported as showing a 57.8% reduction in total OPL area in um2 at P8, p = 0.0286; and Figure 5H is reported as showing the 3.5% of axons that failed to reach the OPL at P8, p = 0.0286. It seems highly unlikely that all these 4 different measurements of different things, at 2 different developmental stages, would all result in p-values that are exactly identical.

2) The authors use 2 main Cre lines to knockout LKB1, one that is widely expressed in the retina, and another that is expressed only in the horizontal cells.

2a) The authors conclude that LKB1 is not acting cell-autonomously in the horizontal cells, because the HC knockout mice do not show the same defects as the pan-retinal KO mice (Figure 4). The authors should demonstrate that LKB1 is in fact gone in the HCs in those mice (possibly by immunostaining for LKB1 to show that there is no protein in the HCs in those mice), before they interpret the absence of a phenotype.

2b) Is LKB1 acting cell autonomously in the cones? Is there a Cre line that would KO LKB1 in the cone cells, to demonstrate where LKB1 is functionally required? Based on the temporal sequence of events, it seems like the cone axon extension defects at P1 precede the P5 HC defects, which along with the lack of a phenotype in the LKB1HC mice, would suggest that the cone cells are driving this process. A new mouse line is beyond the existing resources of the manuscript. However, the Introduction suggests that they'll demonstrate what cells are driving this refinement process, but in the absence of a cone-specific LKB1 KO mouse, they can't really back up that claim.

3) There are some processes that show defects at P5 which are resolved by P8 in LKB1 mutants, but other defects persist at or past P8. Temporary defects include the HC remodeling from radial to planar morphology (Figure 3), and most of the cone axon extension defects (Figure 5). Persistent defects include the OPL disorganization in OPL area per FOV and rod soma localization in the INL (Figure 2), ectopic contacts between HCs and non-terminal parts of cone cells (Figure 6), and VGLUT1 localization in cone somas (Figure 8). Given that the title claims that LKB1 coordinates these processes, please comment on why LKB1 is necessary for some of these processes but not others, and why some of them but not others can eventually be resolved in the absence of LKB1.

4) Please explain how you identified where the patches are in Figure 1B-D. Looking by eye, I see a number of patches, both more apical patches in the control and deeper patches in the LBK1RET, that are not labeled with a magenta arrowhead. How did you determine what met the size/shape criteria to be called a patch?

5) The authors show the mislocalization of VGLUT1 in the Lkb1-Ret KO animals in Figure 8-10, but do not test any other synaptic markers to see if they are also mislocalized. While VGLUT1 may be the only marker that was easily visible at P1, as shown in Figure 7, some of the other markers, including the synapse marker PSD95, are visible at P5, when VGLUT1 mislocalization is still very obvious. Is PSD95 also mislocalized at P5, like VGLUT1, and/or at P8?

Reviewer #3:

The manuscript by Burger et al. examines the role of the serine/threonine kinase Lkb1 in regulating formation of the outer plexiform layer (OPL) in the retina. Previous work from Dr. Samuel identified a role for the Lkb1-AMPK pathway in synaptic remodeling of horizontal cells in the OPL of the aging retina (Samuel et al., 2014). However, it was unknown if Lkb1 function during the development or maintenance of these retinal neurons and their synapses. The present study addresses this by using conditional deletion of Lkb1 and observing the development of cones, horizontal cells, and their synapses throughout their development. Horizontal cells normal remodel their dendrites from an AP polarity to a lateral one between P3 and P5; this process is delayed in Lkb1 mutants. They nicely show that this is a non-cell autonomous effect that arises due to a delay in cone photoreceptor axon extension towards the nascent OPL. Interestingly, horizontal projections extend into the outer nuclear layer and make ectopic contacts on cone cell bodies and stunted axons in the Lkb1 mutants. These ectopic contact sites co-localize with presynaptic Vglut1, suggesting they may identify synapses.

The authors are commended for the quality of their imaging data and the robust quantification methods employed. In addition, the manuscript is very well written and organized, leading to understandable conclusions. The main issue is whether the findings in this paper are novel enough for publication in *eLife*. The horizontal cell phenotypes observed here are largely the same as what the same group previously reported in older animals. There is not really any new information on the upstream or downstream signals related to Lkb1 that provide insight into its function or how the cone axon extension phenotype arises. In addition, Lkb1 has previously been shown to have a role in axon extension, so the delay in cone axon growth in the retina is not entirely unexpected. While the notion that cone axon contacts may be important for horizontal dendrites to remodel from AP to lateral projections is interesting, it is not entirely unprecedented: horizontal dendrite remodeling frequently occurs in response to photoreceptor axon retraction in a number of contexts. Finally, in the Discussion the authors seem to imply that Vglut1 mislocalization (in cones) may play a key role in the horizontal cell phenotypes (subsection “Defective VGLUT1 localization in LKB1 mutant mice”), going as far as to suggest that "Vglut1 may act as a contact guidance cue". However, they do not have any data that directly support this idea. Isn't it just as likely that the ectopic horizontal contacts on cones recruit presynaptic receptors and associated scaffolding proteins that could result in the ectopic Vglut1 localization in cones?

There are a few other points for consideration:

1) Have the authors looked at mosaic arrangement of horizontal cells and cones in the Lkb1 mutants? Do the defects in the Lkb1 mutants lead to uneven axon/dendrite coverage in the OPL plexus?

2) In Figure 3E, can the authors quantify the frequency of ectopic horizontal cell projections in Lkb1 mutants at P8?

3) Lkb1 mutants show a delayed axon extension from the cone photoreceptors; is this just an axon phenotype, or are cone outer segments also delayed in the Lkb1 mutants?

---

## [Author Response]

[Editors’ note: the authors resubmitted a revised version of the paper for consideration. What follows is the authors’ response to the first round of review.]

Summary:As you will see, the reviewers were very interested in role of LKB1 in neurite remodeling in the retina, and the generally high quality of your work. One of the reviewers was generally satisfied with the manuscript. However, the other two reviewers had concerns that focus around the impact of this work compared to your prior paper in 2014; a number of specific technical questions; and concerns with the evidence to support some of your conclusions. In particular, they thought the role of VGlut1 was overstated and would require substantially more evidence, and they were unconvinced by the experiments showing that LKB1 acts non-cell autonomously in horizontal cells. The full original comments of the reviewers are below.

Summary responses to primary concerns:

1) Impact of this work relative to previous data.

Several key distinctions set this work apart from our previous data and that of the field. First, it is impactful because we demonstrate a defined molecular process that links pre and postsynaptic neuron structure to synapse formation. Second, we show that synapse layer emergence differs from synapse decline in three important ways: (i) it involves unique cellular processes; (ii) it employs different neurons and synapses; and (iii) it utilizes distinct downstream molecular pathways. We explain these points below.

a) We directly link neuron structure to synapse formation. The relationship between the structure and location of neuron development and precise synapse formation has remained opaque in most systems. This is because many neurons form a large number of synapses, and these synapses are broadly distributed along the neuron. In the retina, we can precisely interrogate the relationship between presynaptic structure, postsynaptic refinement, and synapse emergence. We use this organization and leverage LKB1 induced changes in neuron shape to show that cone axon extension and synapse protein localization coordinately regulates post-synaptic neuron refinement and synapse emergence.

b) Synapse formation involves district cellular processes. This study focuses on how neurons acquire their paired shapes and connectivity, not how they lose them. We show here that synapse emergence and decline involve fundamentally different cellular attributes and drivers. Horizontal cells provide an excellent example of this difference. In development, these neurons need to arrive, mature, and then transform from a widely dispersed migrating cell into a highly lateralized neuron with a dendrite and an axon. In aging, horizontal cells dendrite are unaffected, while axons that were happily lateralized for ~2 years begin to extend neurites (Samuel et al., 2014). Thus, developmental *refinement* is biologically distinct from the *confinement* defects observed in aging.

c) Synapse layer emergence is driven by different presynaptic neurons than synapse aging. We show here that synapse emergence in retina involves cone-horizontal interactions. In contrast, aging is driven by rods. Cone and rod synapses have vastly different structures and even laminate in distinct regions of the OPL. It has long been a mystery what sets up synaptic organization at this first synapse in vision, and we shed light on that important problem here.

d) Molecular drivers of synapse emergence differ from drivers of synapse aging. In aging, LKB1 functions in rods via AMPK while cones are unaltered (Samuel et al., 2014). We now show that developmental synapse emergence is independent of AMPK (Figure 1—figure supplement 2). These data support the idea that aging cannot simply be viewed as a ‘reversal of development’ even when similar upstream molecules are involved. In our view, this is a strength of our manuscript. It should also be noted that in aging LKB1 signaling does not impact synaptic protein localization, as it does in development. This again supports the idea that the mechanisms differ.

2) Overstating the role of VGLUT1. Thanks for this comment, and we apologize if our data were overinterpreted. This was not our intent. We agree with the reviewers that we cannot conclude from this data alone that VGLUT1 drives this process. We have revised our language accordingly. In addition, since submission we also have collected additional data regarding other synaptic protein defects, which we have included in the revised manuscript.

First, we’ve now shown that other synapse proteins are altered in cone terminal localization in addition to VGLUT1, and generally show low levels in LKB1 mutants relative to controls (Figure 7). Notably, however, unlike VGLUT1, these other synapse proteins are not present at high levels even in control animals until at or after OPL emergence and therefore would be unlikely to drive alterations in cone-horizontal cell contacts.

Second, we have conducted expansion microscopy to highlight differences in levels and location of Ribeye as a representative additional synapse component (Figure 8).

3) Unconvinced of the role of LKB1 in horizontal cells. To confirm that LKB1 is indeed missing from horizontal cells in the LKB1F/F:Cx57iCre animals, we have conducted preliminary RNAscope for LKB1 in this line together with staining for horizontal cells and include this data here (Figure 4—figure supplement 1). These data support that LKB1 transcript levels are decreased in LKB1F/F: Cx57iCre horizontal cells, and we would be happy to provide additional quantitative analysis. Previous papers have also confirmed that Cx57iCre is active by P2, before horizontal cell defects are observed (Barasso et al., 2018).

Reviewer #1:[…] I have 2 questions that I hope the authors can address in the Discussion or Results.1) Do they think (or have evidence for the idea) that horizontal cells are targeting VGLUT1 per se, or their terminals?

Given that VGLUT1 is mislocalized in cone photoreceptors, we quantified the number of terminal and non-terminal contacts that horizontal cells make onto cones. Of those at non-terminal contacts, we then quantified those that contacted mislocalized VGLUT1. Nearly all mistargeted horizontal processes made contacts with ectopic VGLUT1 locations (Figure 10). However, we agree that this is supportive but not direct evidence that VGLUT1 (which is the primary transporter for glutamate in cones) regulates the location of cone-horizontal cell contacts. As noted above, we will be happy to revise our language accordingly. However, since submission we also have collected additional data to bolster the role of VGLUT1 in this process, which we would also be happy to include in a revised manuscript. First, we’ve now shown that other synapse proteins are altered in cone terminal localization in addition to VGLUT1, and generally show low levels in LKB1 mutants relative to controls. Notably, however, unlike VGLUT1, these other synapse proteins are not present at high levels even in control animals until at or after OPL emergence and therefore would be unlikely to drive alterations in cone-horizontal cell contacts (Figure 7). Second, we have conducted expansion microscopy to highlight differences in levels and location of Ribeye as a representative additional synapse component (Figure 8).

2) Is there anything about the sequence or structure of VGLUT1 that makes this a real possibility? Similar mechanisms have, after all, been observed for L-Type Ca^2+^ channels at NMJs.

Thank you for this comment. Indeed, similar mechanisms are involved with L-type calcium channels in synapse formation at the NMJ. Loss of these channels results in loss of acetycholine receptor patterning as well as excessive nerve branching and failure of axons to recognize postsynaptic targets (Kaplan et al., 2018; Kaplan and Flucher, 2019). This will be a good point to address in the Discussion. Given that VGLUT1 is a transporter found on synaptic vesicles, it itself is unlikely directly involved with targeting, but rather glutamate signaling could be a driver of the phenotypes that we observe, just as calcium at the NMJ regulates similar processes.

Reviewer #2:[…] Overall, the work is generally interesting to the field, though its impact could be improved with some additional experiments.1) I am perplexed/alarmed by the many reported p-values of p=0.0286. Specifically Figure 1C is reported as showing a 40.1% reduction in number of patches at P3, p = 0.0286; Figure 2A, B is reported as showing an 85.6% increase in the number of ectopic nuclei at P8, p = 0.0286; Figure 2C, D is reported as showing a 57.8% reduction in total OPL area in um2 at P8, p = 0.0286; and Figure 5H is reported as showing the 3.5% of axons that failed to reach the OPL at P8, p = 0.0286. It seems highly unlikely that all these 4 different measurements of different things, at 2 different developmental stages, would all result in p-values that are exactly identical.

The reviewer is correct that there were several P-values that were similar. We have double checked our numbers and our statistical methods and find the reported numbers to be accurate with one exception, where the correct calculated value is 0.0294. We used the Mann-Whitney Rank-Sum test in Prism6 to determine the significance of differences quantified, given that we cannot assume a normal distribution of the data. We also employed an in-house biostatistician to verify our calculations in R.

2) The authors use 2 main Cre lines to knockout LKB1, one that is widely expressed in the retina, and another that is expressed only in the horizontal cells.2a) The authors conclude that LKB1 is not acting cell-autonomously in the horizontal cells, because the HC knockout mice do not show the same defects as the pan-retinal KO mice (Figure 4). The authors should demonstrate that LKB1 is in fact gone in the HCs in those mice (possibly by immunostaining for LKB1 to show that there is no protein in the HCs in those mice), before they interpret the absence of a phenotype.

Previous papers have confirmed that Cx57iCre is active by P2, before horizontal cell defects are observed (Barasso et al., 2018). To confirm the loss of LKB1 in these cells, we have conducted preliminary RNA-scope for LKB1 in this line together with staining for horizontal cells and include this data here (Figure 4—figure supplement 1). These data suggest that LKB1 transcripts are absent from horizontal cells in these mice. Unfortunately, IHC for LKB1 is not possible due to lack of compatible antibodies (we’ve tested five so far). We would also be happy to perform a time course analysis of Cx57iCre; Ai14 expression to show when the Cre is active and how many cells it effects.

2b) Is LKB1 acting cell autonomously in the cones? Is there a Cre line that would KO LKB1 in the cone cells, to demonstrate where LKB1 is functionally required? Based on the temporal sequence of events, it seems like the cone axon extension defects at P1 precede the P5 HC defects, which along with the lack of a phenotype in the LKB1HC mice, would suggest that the cone cells are driving this process. A new mouse line is beyond the existing resources of the manuscript. However, the Introduction suggests that they'll demonstrate what cells are driving this refinement process, but in the absence of a cone-specific LKB1 KO mouse, they can't really back up that claim.

Deleting LKB1 from cones early in development is a challenge given the available resources. We have considered 4 approaches in good faith but none are suitable. The first is a mouse line. The only Cone-specific cre that has been widely used is HRGP-Cre. We have this line, but unfortunately the Cre is not active until P8, after OPL emergence has occurred. A second approach is electroporation, but this too spares cones because cones are done dividing by birth and therefore do not receive the constructs (Sarin et al., 2018). Third, we have tried electroporation at E15 but find that GFP dilutes out in subsequent cell divisions, making it difficult to discern what cells our electroporation targeted. Finally, we have considered injecting AAV. However, given that AAV takes 21 days for full expression we are unable to visualize transduction by P5 even if viruses are injected early.

3) There are some processes that show defects at P5 which are resolved by P8 in LKB1 mutants, but other defects persist at or past P8. Temporary defects include the HC remodeling from radial to planar morphology (Figure 3), and most of the cone axon extension defects (Figure 5). Persistent defects include the OPL disorganization in OPL area per FOV and rod soma localization in the INL (Figure 2), ectopic contacts between HCs and non-terminal parts of cone cells (Figure 6), and VGLUT1 localization in cone somas (Figure 8). Given that the title claims that LKB1 coordinates these processes, please comment on why LKB1 is necessary for some of these processes but not others, and why some of them but not others can eventually be resolved in the absence of LKB1.

We thank the reviewer for these comments. While many defects are still present at P8, cone axons have largely extended (though their final length is shorter than controls) so there appear to be some redundant mechanisms at play here. However, it is worth noting that this does not mean that the circuit is functioning normally. Indeed, we’ve shown that adult LKB1 mutants display defective rod and cone visual signaling (Samuel et al., 2014).

4) Please explain how you identified where the patches are in Figure 1B-D. Looking by eye, I see a number of patches, both more apical patches in the control and deeper patches in the LBK1RET, that are not labeled with a magenta arrowhead. How did you determine what met the size/shape criteria to be called a patch?

We appreciate this suggestion, and we would be happy to clarify how patches are delineated. To define a patch, we asked whether a gap in the nuclear stain overlapped with the location of either a horizontal cell terminal, a cone terminal, or both, since the neurites of these cells are first to form contacts in development. The gaps to which the reviewers refer are caused by horizontal cell soma and do not reflect points of contact. We have now clarified how patches were identified and quantified in the text (subsections “Cone axon extension and maturation are disrupted in LKB1 mutant mice” and “LKB1 is required for early localization of VGLUT1).

5) The authors show the mislocalization of VGLUT1 in the LKB1RET KO animals in Figure 8-10, but do not test any other synaptic markers to see if they are also mislocalized. While VGLUT1 may be the only marker that was easily visible at P1, as shown in Figure 7, some of the other markers, including the synapse marker PSD95, are visible at P5, when VGLUT1 mislocalization is still very obvious. Is PSD95 also mislocalized at P5, like VGLUT1, and/or at P8?

We have included in this revised manuscript additional data on other synapse proteins in LKB1 mutants (Figure 7). We show that other synapse proteins are indeed altered in their localization to the cone terminal. To illustrate this, we have completed additional expansion microscopy showing differences in levels and location of Ribeye (Figure 8). Notably, however, unlike VGLUT1, these other synapse proteins are present at or after OPL emergence so are unlikely to contribute to OPL formation.

Reviewer #3:[…] The authors are commended for the quality of their imaging data and the robust quantification methods employed. In addition, the manuscript is very well written and organized, leading to understandable conclusions. The main issue is whether the findings in this paper are novel enough for publication in eLife. The horizontal cell phenotypes observed here are largely the same as what the same group previously reported in older animals. There is not really any new information on the upstream or downstream signals related to Lkb1 that provide insight into its function or how the cone axon extension phenotype arises. In addition, Lkb1 has previously been shown to have a role in axon extension, so the delay in cone axon growth in the retina is not entirely unexpected. While the notion that cone axon contacts may be important for horizontal dendrites to remodel from AP to lateral projections is interesting, it is not entirely unprecedented: horizontal dendrite remodeling frequently occurs in response to photoreceptor axon retraction in a number of contexts.

We thank the reviewer for these comments and apologize that we did not make the distinction clearer between our previous work and this manuscript. In fact, they are different at the cellular, synaptic, and molecular level. We have clarified these points in the section above entitled “Impact of this work relative to previous data”.

Finally, in the Discussion the authors seem to imply that Vglut1 mislocalization (in cones) may play a key role in the horizontal cell phenotypes (subsection “Defective VGLUT1 localization in LKB1 mutant mice”), going as far as to suggest that "Vglut1 may act as a contact guidance cue". However, they do not have any data that directly support this idea. Isn't it just as likely that the ectopic horizontal contacts on cones recruit presynaptic receptors and associated scaffolding proteins that could result in the ectopic Vglut1 localization in cones?

Please refer to point 3 in our general responses above. It was not our intent to overstate our findings and agree with the reviewers that we cannot conclude from this data alone that VGLUT1 drives this process (though it may indeed contribute). We will be happy to revise our language accordingly. However, since submission we also have collected additional data, which we would be happy to include in a revised manuscript. First, we’ve now shown that other synapse proteins are altered in cone terminal localization in addition to VGLUT1, and generally show low levels in LKB1 mutants relative to controls. Notably, however, unlike VGLUT1, these other synapse proteins are not present at high levels even in control animals until at or after OPL emergence and therefore would be unlikely to drive alterations in cone- horizontal cell contacts (Figure 7). Second, we have conducted expansion microscopy to highlight differences in levels and location of Ribeye as a representative additional synapse component (Figure 8).

There are a few other points for consideration:1) Have the authors looked at mosaic arrangement of horizontal cells and cones in the Lkb1 mutants? Do the defects in the Lkb1 mutants lead to uneven axon/dendrite coverage in the OPL plexus?

Thank you for this suggestion. We have examined mosaic spacing in the LKB1 mutants and see no difference in spacing or OPL coverage (Author response image 1).

2) In Figure 3E, can the authors quantify the frequency of ectopic horizontal cell projections in Lkb1 mutants at P8?Thank you for this suggestion. We have provided quantifications as Author response image 2 but would be happy to include this data as a supplementary figure if deemed necessary.

**Author response image 2. respfig2:** 

3) Lkb1 mutants show a delayed axon extension from the cone photoreceptors; is this just an axon phenotype, or are cone outer segments also delayed in the Lkb1 mutants?Cone outer segments appear normal during development. They can be observed in Figure 5.